# Wood hemicelluloses exert distinct biomechanical contributions to cellulose fibrillar networks

Jennie Berglund[1], Deirdre Mikkelsen [2,3], Bernadine M. Flanagan[2], Sushil Dhital[2,4], Stefan Gaunitz[1,5], Gunnar Henriksson[1], Mikael E. Lindström[1], Gleb E. Yakubov[6,7✉], Michael J. Gidley [2✉] & Francisco Vilaplana [1,5✉]

Hemicelluloses, a family of heterogeneous polysaccharides with complex molecular structures, constitute a fundamental component of lignocellulosic biomass. However, the contribution of each hemicellulose type to the mechanical properties of secondary plant cell walls remains elusive. Here we homogeneously incorporate different combinations of extracted and purified hemicelluloses (xylans and glucomannans) from softwood and hardwood species into self-assembled networks during cellulose biosynthesis in a bacterial model, without altering the morphology and the crystallinity of the cellulose bundles. These composite hydrogels can be therefore envisioned as models of secondary plant cell walls prior to lignification. The incorporated hemicelluloses exhibit both a rigid phase having close interactions with cellulose, together with a flexible phase contributing to the multiscale architecture of the bacterial cellulose hydrogels. The wood hemicelluloses exhibit distinct biomechanical contributions, with glucomannans increasing the elastic modulus in compression, and xylans contributing to a dramatic increase of the elongation at break under tension. These diverging effects cannot be explained solely from the nature of their direct interactions with cellulose, but can be related to the distinct molecular structure of wood xylans and mannans, the multiphase architecture of the hydrogels and the aggregative effects amongst hemicellulose-coated fibrils. Our study contributes to understanding the specific roles of wood xylans and glucomannans in the biomechanical integrity of secondary cell walls in tension and compression and has significance for the development of lignocellulosic materials with controlled assembly and tailored mechanical properties.

---

[1] Wallenberg Wood Science Centre, Department of Fibre and Polymer Technology, School of Engineering Sciences in Chemistry, Biotechnology and Health, KTH Royal Institute of Technology, Stockholm, Sweden. [2] ARC Centre of Excellence in Plant Cell Walls, Centre for Nutrition and Food Sciences, Queensland Alliance for Agriculture and Food Innovation, The University of Queensland, St. Lucia, Brisbane, QLD 4072, Australia. [3] School of Agriculture and Food Sciences, The University of Queensland, St. Lucia, Brisbane, QLD 4072, Australia. [4] Department of Chemical Engineering, Monash University, Clayton, VIC 3800, Australia. [5] Division of Glycoscience, Department of Chemistry, School of Engineering Sciences in Chemistry, Biotechnology and Health, AlbaNova University Centre, KTH Royal Institute of Technology, Stockholm, Sweden. [6] ARC Centre of Excellence in Plant Cell Walls, School of Chemical Engineering, The University of Queensland, St. Lucia, Brisbane, QLD 4072, Australia. [7] Faculty of Sciences, School of Biosciences, University of Nottingham, Nottingham, UK. ✉email: gleb.yakubov@nottingham.ac.uk; m.gidley@uq.edu.au; franvila@kth.se

Plant cell walls are dynamic polymeric networks, which confer structural stability to the cells, protection against pathogens, permeability, and regulatory functions during growth[1]. Plant cell walls exhibit a layered organization in primary and secondary walls, where the composition, molecular structure, and assembly in each layer confer specific properties to the overall hierarchical cell wall architecture. Primary cell walls are synthesized during cell growth and have a highly hydrated architecture consisting of non-oriented cellulose microfibrils interlocked by hemicelluloses, pectins, and structural proteins. Primary cell walls must show both elastic and plastic biomechanical performance to withstand cell pressure and enable cell extension[2,3]. On the other hand, secondary cell walls show a fully differentiated structure in multiple lamellae, with a larger content of oriented cellulose microfibrils embedded in a matrix of hemicelluloses and lignin, providing compressive and tensile strength, toughness, and rigidity to the woody cells after expansion[2,3]. The biomechanical integrity of secondary cell walls is regulated by the molecular interactions in specific nanodomains between cellulosic microfibrils, hemicelluloses, and lignins, resulting in distinct interfacial areas (cellulose with adsorbed hemicelluloses, hydrated hemicellulose domains, and lignin–carbohydrate complexes). However, the chemical nature and structural impact of the interactions between secondary cell wall components in the interfacial areas is still controversial[4–6]. The crystalline structure and the orientation of cellulose fibers (the so-called microfibril angle) contribute largely to the stiffness of the secondary cell walls[2]. Lignification, on the other hand, provides rigidity and hydrophobicity, preventing the swelling of secondary walls in water, which is fundamental for water conducting requirements in vascular tissues. Finally, hemicelluloses have been suggested to act as a link between the lignin and cellulose components in the cell wall, and to regulate the aggregation of cellulose microfibrils[7]. Hemicelluloses are a family of complex biopolymers sharing with cellulose a β-(1→4) backbone of neutral sugars (glucose, xylose, and mannose), but decorated with an array of neutral sugar and uronic acid substitutions and can be chemically modified by acetylation. The molecular heterogeneity of hemicelluloses is key in modulating the interactions with cellulose microfibrils through intermolecular interactions[8–10] and the occurrence of covalent linkages with lignin[11–13]. Indeed, the hemicellulose xylan changes its conformation when adsorbed onto cellulose surfaces[9,10,14], using specific motifs that may fine-tune the supramolecular interactions with cellulose surfaces by hydrogen bonding and non-polar forces[8,15]. However, the contribution of the different hemicelluloses and their molecular motifs to the biomechanical integrity and recalcitrance of lignocellulose materials is far from understood.

Hemicelluloses represent one-third of the total lignocellulosic biomass and are of major interest due to their potential use in bio-based products such as barrier films[16,17] and additives in food and pharmaceutical products[18]. The technical utilization of hemicelluloses is limited due to their thermo-chemical degradation during extraction and downstream processing[19], and the lack of purity and structural control over the isolated fractions. Another limiting factor is our poor understanding of the effect of hemicellulose on the mechanical and viscoelastic properties of lignocellulose-based materials. This is hindered by the lack of available models and techniques where the individual contribution of specific hemicellulose features can be assessed in planta. In this work, secondary cell wall hemicelluloses from birch (*Betula pendula* and *Betula pubescens*) and Norway spruce (*Picea abies*) as models for hardwood and softwood species, respectively, have been extracted and purified, and incorporated in bacterial cellulose (BC) hydrogels (Fig. 1a). The major hemicellulose in birch is *O*-acetyl-4-*O*-methylglucuronoxylan (acGX); in spruce, on the other hand, *O*-acetyl-galactoglucomannan (acGGM) is the most common hemicellulose (20% of total weight) and arabino-4-*O*-methylglucuronoxylan (AGX) is also significantly present (10% of total weight) (Fig. 1b)[20–22].

Bacterial systems that secrete cellulose extracellularly constitute useful models for studying plant cell wall assembly and the properties of polysaccharide networks, as the composition and molecular structure of the hydrogels can be tailored by adding specific polysaccharide components to the medium during fermentation. BC has a larger proportion of the $I_\alpha$ crystalline allomorph, whereas higher plants have a higher content of the $I_\beta$ allomorph[23]. However, the surface properties and fundamental aspects of hemicellulose adsorption should remain undeterred by the subtle differences in the crystalline arrangement between the bacterial and plant cellulose systems. Indeed, BC hydrogels have been used to study interactions with primary cell wall polysaccharides such as xyloglucan[4,24], pectins[25,26], arabinoxylans[27] and mixed-linkage β-glucans[28], seed galacto- and glucomannans[29], and wood hemicelluloses[30–32]. However, the effect of purified and well-defined wood hemicellulose compositions on the tensile and compression biomechanical properties of multicomponent cellulose hydrogels has not been studied before.

We herein prepare multicomponent BC hydrogels with controlled wood hemicellulose compositions mimicking the secondary cell wall of hardwoods and softwoods prior to lignification, to identify the contributions of wood mannans and xylans to the biomechanical properties of cellulose fibrillar networks. Wood glucomannans increase the compression properties of the cellulose fibrillar networks, whereas xylans increase the ductility under tension. These distinct biomechanical contributions are related to the specific molecular structures of the hemicelluloses and to their organization in both rigid and flexible phases, which contribute to the multiscale architecture of the BC hydrogels. The study of these multiscale interactions provides a framework for understanding the assembly, biomechanical properties, and biological function of secondary plant cell walls, and contributes to the development of cellulose-based materials and products.

## Results

**Wood hemicelluloses are homogenously incorporated into BC networks.** Different hemicelluloses from birch (*B. pendula* and *B. pubescens*) and spruce (*P. abies*) representing soft- and hardwoods, respectively, were extracted and purified using both alkaline and hydrothermal processes (Supplementary Fig. 1a). A set of wood hemicelluloses with distinct backbone and decoration structures were thus obtained (Fig. 1b), including AGX and acGGM from spruce, a mixture of AGX and deacetylated GGM from spruce (AGX + GGM$_{alk}$), and acGX from birch. The obtained hemicellulose fractions were characterized in terms of their average monosaccharide composition, acetyl content, and molar mass (Supplementary Table 1). Extracted hemicelluloses might contain lignin impurities[13,33]. However, no significant lignin resonances were observed from the two-dimensional nuclear magnetic resonance (2D-NMR) analysis (Supplementary Fig. 2), and the contribution from possible lignin as well as pectin substances indicated from the galacturonic acid (GalA) was negligible. The intramolecular substitution patterns of the extracted hemicelluloses were analyzed in detail using enzymatic hydrolysis with selective glycosyl hydrolases and subsequent identification of the released oligosaccharide motifs (Fig. 1c). A specific β-glucuronoxylanase (GH30) releasing acidic oligosaccharides was used for the analysis of wood xylans, which enables the identification of the glucuronic acid spacing[10,34,35]; on the other hand, a β-mannanase (GH5) was selected for the

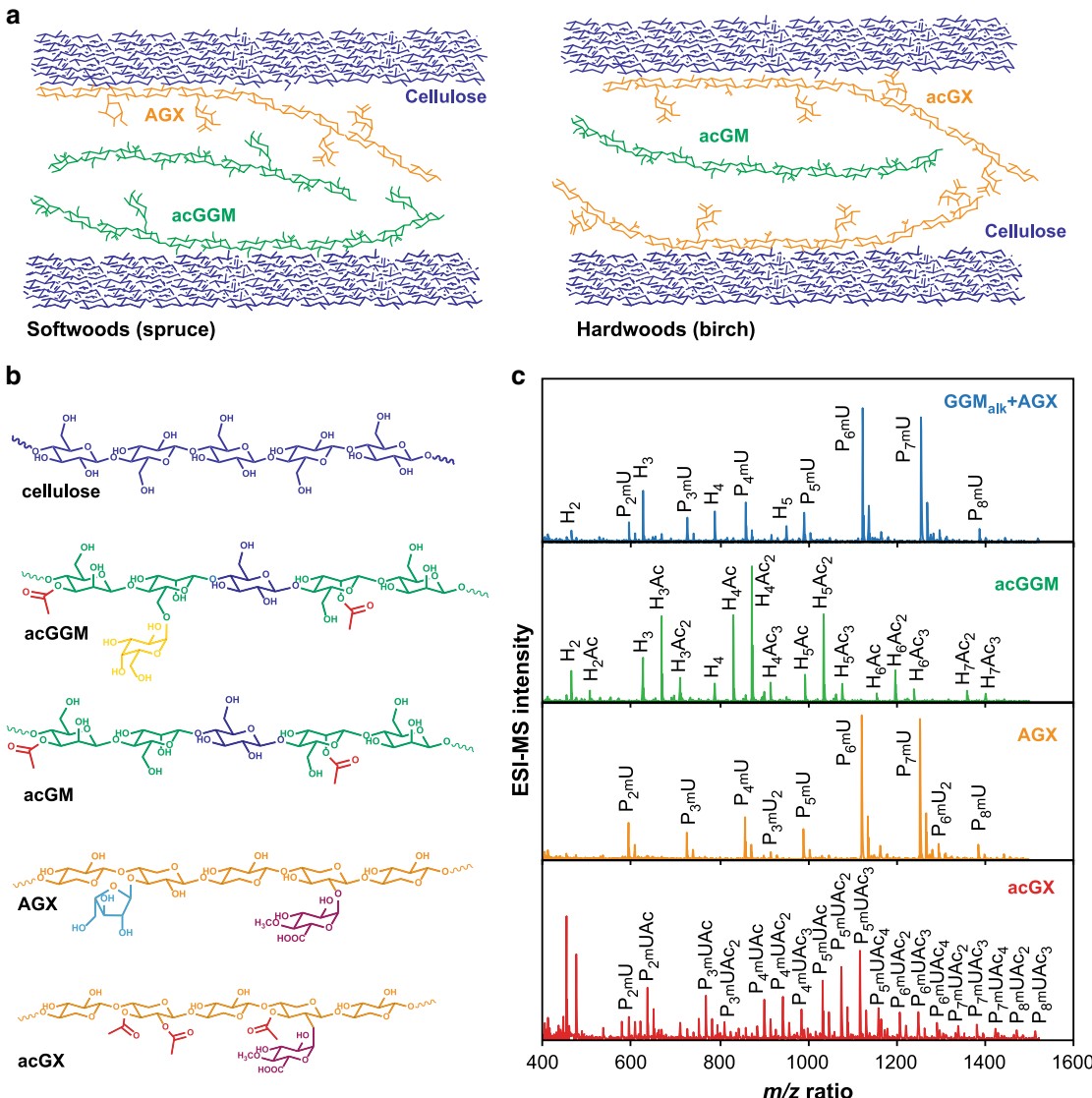

**Fig. 1 Structure of wood hemicelluloses and their incorporation in BC-H model networks. a** Schematic illustration of the polymeric network in a model BC-H system mimicking the secondary cell walls of softwoods and hardwoods prior to lignification. **b** Average molecular structure of major repeating motifs of wood hemicelluloses. Softwood O-acetyl-galactoglucomannan (acGGM, from spruce) comprises a main chain of β-(1→4)-linked D-mannose (Man) and D-glucose units (Glc) partially substituted by α-(1→6)-linked D-galactose units (Gal) exclusively on the Man units and partial acetylation at the C2 and C3 hydroxyl groups of Man[68]. Hardwood O-acetyl-glucomannan (acGM) has similar structure as acGGM without Gal substitutions. Softwood arabino-4-O-methylglucuronoxylan (AGX, from spruce) consists of a backbone of β-(1→4)-linked D-Xyl with partial substitution by both α-(1→2)-linked mGlcA and α-(1→3)-linked L-arabinofuranose (Ara)[10,16]. Hardwood O-acetyl-4-O-methylglucuronoxylan (acGX, from birch) consists of a backbone of β-(1→4)-linked D-xylose units (Xyl) with partial substitution of α-(1→2)-linked 4-O-D-methylglucuronic acids (mGlcA) and O-acetylation (OAc) at some of the C2 and C3 units of Xyl[13,22]. **c** Oligomeric mass profiling (OLIMP) by mass spectrometry of the wood hemicelluloses (AGX, acGX, and acGGM) after enzymatic hydrolysis. Note: P refers to a pentose (Xyl or Ara), H refers to a hexose (Man, Gal or Glc), Ac refers to an acetyl group, and mU refers to mGlcA. Source data provided as a Source Data file.

enzymatic profiling of spruce mannans[36]. Birch acGX was extracted directly from acetone-washed wood using subcritical water without previous delignification and the extracts were further purified by ethanol precipitation to remove residual lignin. Birch acGX has a complex pattern of glucuronidation and acetylation, with a distribution of xylo-oligosaccharides with mGlcA spacing every three to eight Xyl units carrying one to four acetyl groups. In our previous work, we reported the correlation between acetylation and glucuronidation in acGX from birch wood[13], with regularly spaced acetyl groups in the Xyl units carrying a mGlcA side group and further evenly spaced 2-O and 3-O-acetylations. This regularity in the acetylation patterns has been previously observed for *Arabidopsis thaliana* and has a

functional role in modulating the association with cellulose fibers[8,14]. The alkaline extracted spruce hemicelluloses (AGX + GGM_alk) consisted of about 27 mol% GGM and 68 mol% AGX, evidencing that KOH extraction is more selective towards AGX. The oligomeric mass profiling from the mixture shows oligosaccharide signals arising both from the deacetylated GGM (GGM_alk) as a hexose ladder and the acidic xylo-oligosaccharides from spruce AGX. Spruce AGX was further purified with Ba (OH)₂ and showed the presence of major domains with regular placement of substitutions along the backbone (with glucuronidation every six Xyl units and Ara substitutions evenly spaced with respect to the mGlcA), and minor domains with odd and consecutive glucuronidation spacing, as we have previously

reported[10]. Finally, the water-extracted spruce acGGM shows a complex pattern of acetylated manno-oligosaccharides with no reported regularity up to this point in the position of the acetylation or galactosylation with respect to the mannan backbone[37]. The differences in the distribution of acetyl and glycosyl substitutions between the wood mannans and xylans may play a significant role in the supramolecular assembly of secondary cell walls, regulating the interactions with cellulose and lignin. The hemicelluloses displayed average molar masses of 20–40 kDa (Supplementary Table 1), which correspond approximately with a degree of polymerization of 100–200 units. Assuming a length of 1 nm for a cellobiose unit, this provides a theoretical maximum length of 50–100 nm for a hemicellulose chain in a strictly rigid elongated conformation. However, hemicelluloses adopt coiled conformations with hydrodynamic radius in the order of 10 nm when molecularly dissolved in water and have a large tendency to self-aggregate in hydrated conditions into larger fractal aggregates that can reach 100–1000 nm[38,39]. These coiled and elongated conformations can be envisioned as two extreme cases, as individual hemicellulose macromolecules can adopt different conformations when adsorbed onto cellulose microfibril surfaces and in hydrated states[8,10,14,40].

The well-defined wood hemicelluloses were used for the production of BC–hemicellulose (BC-H) hydrogels with different contents, mimicking the composition of secondary plant cell walls prior to lignification (Supplementary Fig. 1b). Incubation with *Komagataeibacter xylinus* resulted in the production of highly hydrated disk-shaped BC pellicles with a water content of 98–99 wt%, also referred to as hydrogels, on the surface of the Hestrin–Schramm (HS) fermentation medium. The hydrogel model for hardwoods was fabricated by incubation of *K. xylinus* in the presence of acetylated glucuronoxylan from birch (BC-acGX). For the softwood models, separate hydrogels containing each of the two common softwood hemicelluloses (BC-acGGM and BC-AGX) were prepared. Two additional multicomponent BC hydrogels were fabricated containing both softwood hemicelluloses in acetylated and non-acetylated forms (BC-acGGM + AGX and BC-GGM$_{alk}$ + AGX), mimicking the relative abundance of mannans and xylans in spruce wood (ratio of 2 : 1). The solution-state $^1$H NMR of the extracted hemicelluloses and washed out polysaccharides after BC-H harvest showed similar sugar ratio (Supplementary Fig. 3), which indicates that the composition of the hemicelluloses in the BC-H hydrogels corresponds to the extracted hemicelluloses, and suggests that no preferential incorporation of hemicelluloses with specific motifs occurred during the preparation of the hydrogels. The degree of incorporation of hemicelluloses in the BC-H was evaluated after sulfuric acid hydrolysis of the dried BC-H composites, indicating that the hemicellulose content typically ranged between 16 and 27 mol% (Fig. 2a). The high content of birch acGX (27 mol%) might be explained by a higher solubility or stability of the solution during BC production. The monosaccharide analysis of compressed materials used for tensile testing also confirmed that the monosaccharide composition was retained after compression for increased solid content (Supplementary Fig. 4). The incorporated hemicellulose content is slightly lower in our BC-H composites than in the typical secondary cell walls of woody tissues, where the hemicellulose content normally reaches around 30% considering all cell wall components. However, the actual differences between the hemicellulose content in our bacterial BC-H hydrogels and in plant secondary cell walls are not that far off, and the observed effects that we here report are likely to be even more pronounced for higher hemicellulose concentrations.

Post harvest, the successful incorporation of hemicellulose into the cellulose network was confirmed by antibody labeling by LM11 (xylan, represented in red) and LM21 (mannan, represented in green). The hemicelluloses appear to be homogeneously distributed in the BC-H hydrogels, as can be observed from the 2D (Fig. 2b and Supplementary Fig. 5) and the three-dimensional visualizations (Supplementary Fig. 6) of the antibody-labeled BC-H hydrogels under the fluorescent microscope. The microstructure of the BC-H hydrogels was analyzed by scanning electron microscopy (SEM) and compared to the reference BC system (Fig. 2c). The images of the hydrated hydrogels taken from the top of the materials show that all BC-H hydrogels consisted of an apparently random network of fibrillar structures with no obvious orientation at the scale dimensions of the micrographs. The reference BC hydrogel shows the typical ribbon-like structures previously reported with widths around 35–50 nm arising from the bundling of individual cellulose microfibrils[30,41]. Although no directional orientation was observed in the BC hydrogels, some contact points between ribbons could be identified in the reference system. The addition of wood hemicelluloses with controlled molecular structure and composition did not cause a significant effect on the width of the cellulose ribbons or on their orientation as observed from SEM analysis. However, increased coalescence of the cellulose bundles could be observed in the BC-H images, together with some minor nodular structures that could arise from the self-aggregation of the introduced hemicelluloses in the hydrogels. In particular, these nodules could be mainly observed in the BC-H hydrogels containing acGGM (both alone and together with AGX). These observations indicate that the pure wood xylans and mannans were homogeneously incorporated in the BC-H hydrogels, both on the surfaces of the cellulose bundles contributing to their cooperative coalescence, but also finely dispersed between the bundles. This suggests that the BC-H hydrogels are assembled as multiphase systems, where hemicelluloses are both integrated within the cellulose bundles and are also dispersed in the surrounding matrix. This binding effect on the BC ribbons has been previously reported for tamarind seed xyloglucan and low substitution galacto(gluco)mannans, indicating a tight association between both components[28,42], whereas the presence of large nodules not associated with the cellulose fibrils was observed for BC hydrogels containing wheat-endosperm arabinoxylans[28].

**Wood hemicelluloses influence the compression properties of BC-H hydrogels.** The mechanical properties in compression are important for the biomechanical response of secondary plant cell walls and are also significant for formulating lignocellulosic materials and products. Here, the compression behavior of BC-H composite hydrogels was evaluated using a compression–relaxation technique adapted for a rotational rheometer[43], and compared with pure BC hydrogels. The starting concentration of polysaccharides was around 1–2 wt%, with the remainder (98–99 wt%) being water. Such dilute (as compared to the actual plant cell wall) systems make it possible to specifically probe contributions from the polysaccharide network architecture and how the network density—which increases with compression—influences the mechanical response. During the compression cycle, the initial rise in stress in response to the applied compression strain originates from the pressure of the fluid confined within hydrogel pores (poroelastic effect). Upon cessation of compression (i.e., under constant compression strain), the stress quickly relaxes due to fluid drainage through the pores. Figure 3a illustrates a typical set of compression–relaxation profiles recorded using pure BC as an example. For each cycle, the values of the compression stress ($\sigma_x$) and strain ($\varepsilon_x$) describe the effective mechanical response of composite hydrogels (Supplementary Fig. 7). The direct analysis of the compression–relaxation curves without applying any

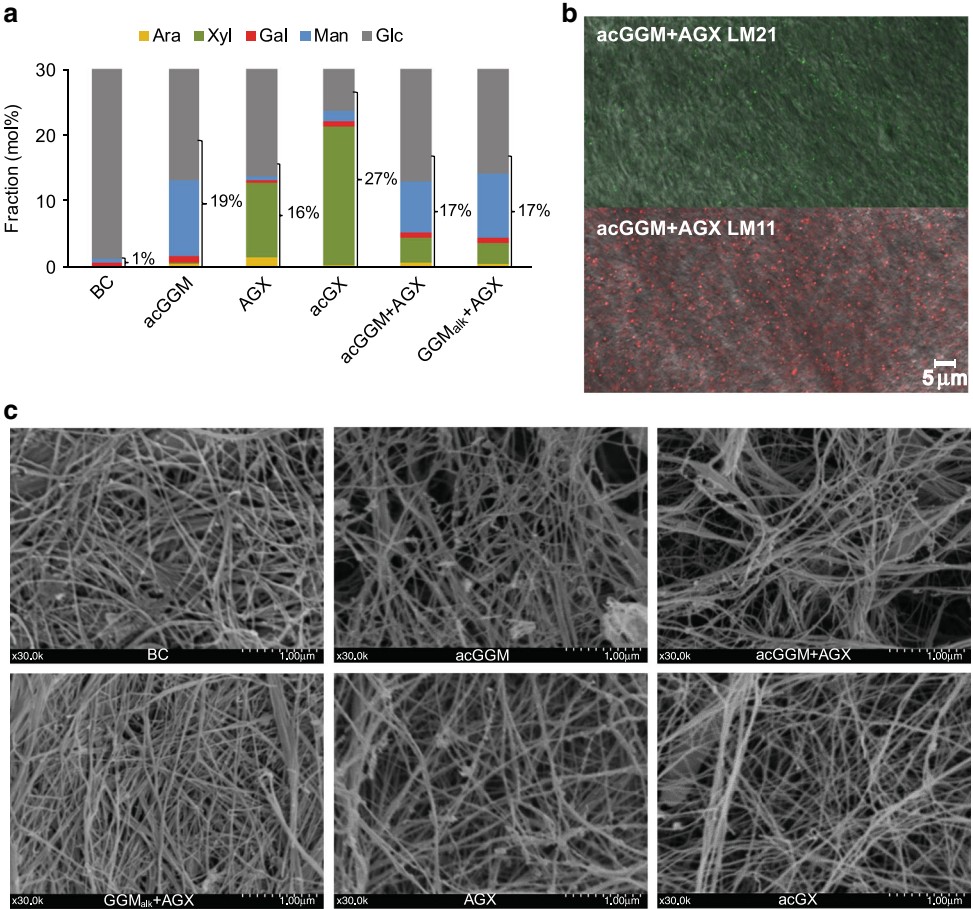

**Fig. 2 Composition and microstructure of the BC-H hydrogels.** The sample names for the BC-H materials correspond to the hemicellulose content. **a** Monosaccharide composition of BC-H composite pellicles. The $y$ axis ends at 30% to amplify the composition of hemicellulose sugars; however, the glucose (Glc) fraction continues up to 100%. SDs: ±0.0–0.8. Values correspond to the fraction of hemicellulose-related sugars, where the Glc contribution from hemicelluloses as well as uronic acid composition was estimated by assuming the same ratio to mannose (Man) or xylose (Xyl) as in the extracted hemicellulose. Source data are provided as a Source Data file. **b** Confocal microscopy images of antibody-labeled BC-acGGM+AGX. Xylans are labeled by LM11 (in red) and mannans are labeled by LM21 (in green). The length of the scale bar corresponds to 5 μm. **c** Scanning electron microscopy (SEM) images of BC-H pellicles (at ×30 magnification). The length of the scale bar corresponds to 1 μm.

specific model enables determination of the compression ratio (CR) and the elastic modulus after relaxation ($E_{relax}$). The CR is directly calculated from the difference in pellicle thickness ($h$) at each compression profile compared to the starting thickness ($h_0^*$) (Eq. 1). $E_{relax}$ is computed using the geometric area of the pellicle (A), the compression strain at each cycle ($\varepsilon_{x, relax}$) (Eq. 2), and the difference in the normal force during the compression step at each cycle ($\Delta F_{x, relax}$) (Eq. 3). The Poisson ratio in BC hydrogels is very close to zero[41,43]. After each relaxation step, a small amplitude oscillatory shear test was performed in the linear viscoelastic region (as determined by the stress sweep data in Supplementary Fig. 8), to determine the material shear moduli ($G'$ and $G''$). The storage modulus $G'$ (showing the elastic contribution) is found to be always larger than the loss modulus $G''$ (viscous contribution) for all relaxation steps (Supplementary Fig. 7). This shows that the BC-H hydrogels exhibit predominantly elastic behavior in agreement with previous works[43].

$$CR_x = \frac{h_0^* - h_0^x}{h_0^*} \quad (1)$$

$$\varepsilon_x = \frac{h_0^{x+1} - h_1^x}{h_0^{x+1}} \quad (2)$$

$$E_{relax}^x = \frac{\Delta F_x / A}{\varepsilon_x} \quad (3)$$

All BC-H composites showed an increase in stiffness with the increase of the CR, as indicated by the increase in $E_{relax}$ and the storage ($G'$) and loss ($G''$) components of the shear modulus (Fig. 3b and Supplementary Fig. 7). At CRs of around 0.1, the differences between the different BC-H hydrogels was found to be most notable. We use this empirical finding to compare the effect of the different hemicelluloses on the viscoelastic behavior under compression and relaxation (Table 1). BC-acGGM showed a higher $E_{relax}$ than the reference BC, whereas $E_{relax}$ for BC-AGX was significantly lower. The combined BC-acGGM + AGX and BC-GGM$_{alk}$ + AGX composites behaved similarly in compression and showed a slightly lower $E_{relax}$ compared to pure BC, but higher than BC-AGX. The composites with BC-acGX demonstrated similar behavior to BC-H composites fabricated using combined spruce hemicelluloses. At the highest values of CR, the measured differences between different materials become less pronounced. This, we hypothesize, is caused by the newly formed adhesive links between cellulose fibrils, which assemble in the

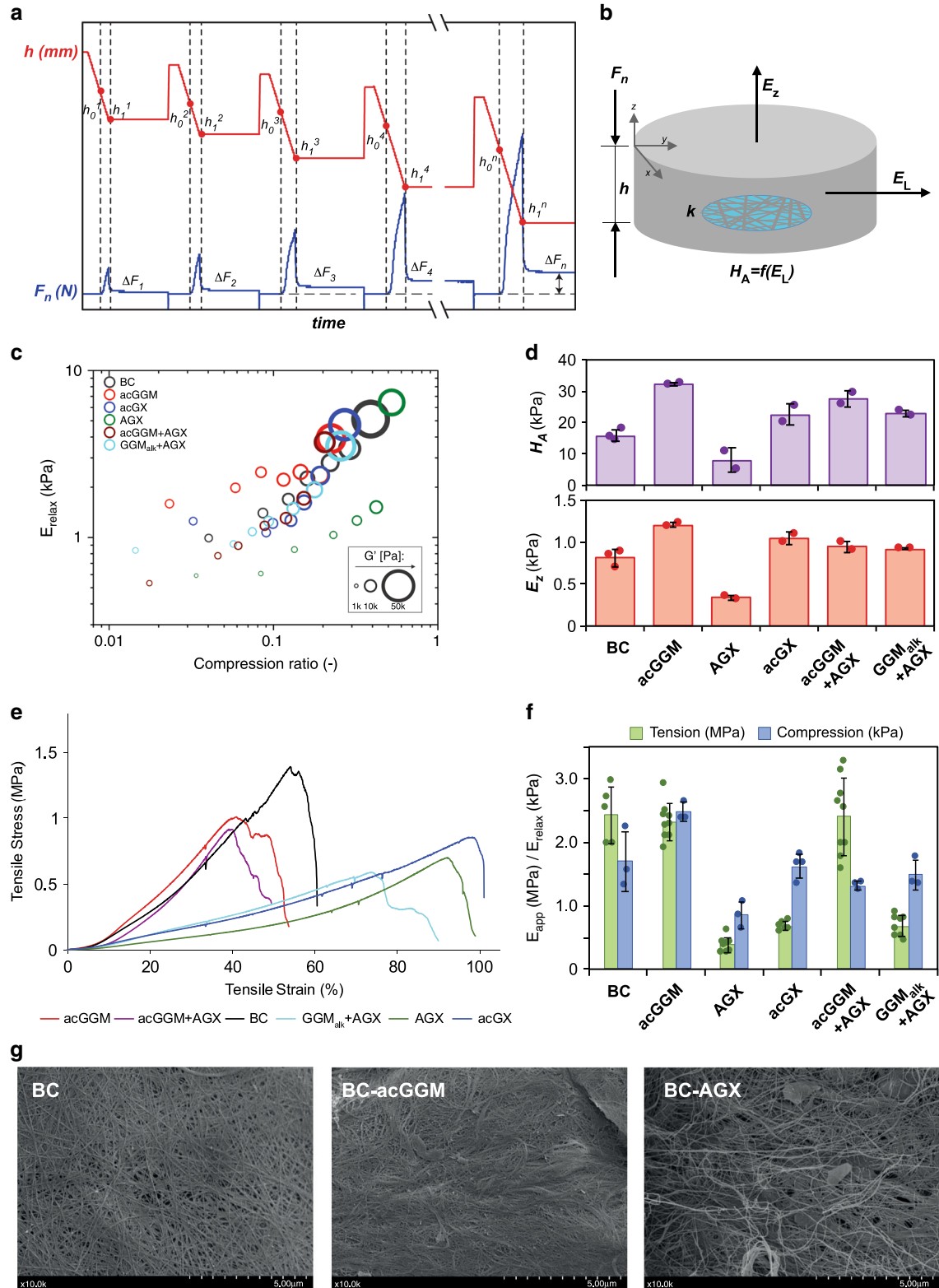

absence of freely available hemicelluloses[44]. Overall, the combined spruce hemicellulose materials and the birch BC-acGX showed similar mechanical response under compression to that of pure BC. In contrast, BC-acGGM and BC-AGX composites showed distinct behavior with the trend towards stiffening and more compliant behavior, respectively. A qualitatively similar behavior was observed for the shear moduli ($G'$ and $G''$) for the different BC-H hydrogels. The introduction of spruce

glucomannan in the hydrogels (BC-acGGM) resulted in an increase of the shear moduli compared to the reference BC hydrogel, which was even more pronounced for the deacetylated glucomannan (BC-deacGGM). On the other hand, spruce xylan induced a significant decrease in the shear moduli; however, this behavior was not replicated for the birch xylan equivalent. The combined softwood hydrogels (BC-acGGM + AGX and BC-GGM$_{alk}$ + AGX) again showed a very similar viscoelastic

**Fig. 3 Mechanical properties of the BC-H hydrogels under compression and tension.** The sample names for the BC-H materials correspond to the incorporated hemicelluloses. **a** Graphical scheme of the compression–relaxation mechanical analysis, where $F_n(N)$ is the normal force and $h$ is the pellicle thickness. **b** Parameters for the poroelastic model in compression, where $E_z$ is the out-of-plane modulus; $H_A$ is the aggregate modulus, which is a function of the lateral (in-plane) modulus ($H_A = f(E_L)$) (see Supplementary Methods, Supplementary Eq. 1); and $k$ is the permeability. **c** Compression–relaxation mechanical analyses. Values are averages from three replicates. Source data are provided as a Source Data file. **d** Moduli obtained from the poroelastic model (at a compression rate of 0.1). Error bars are based on SDs from two to three measurements. **e** Representative stress–strain curves from the tensile testing of BC-H hydrogels after compression. **f** Comparison between the E-modulus in tension ($E_{app}$) and compression ($E_{relax}$ at a compression rate of 0.1). Error bars correspond to SDs ($n = 5$–9 for tension and $n = 3$–4 for compression measurements). **g** Scanning electron microscopy (SEM) images of selected BC-H hydrogels after compression (at ×10 magnification). The length of the scale bar corresponds to 5 µm.

**Table 1 Biomechanical properties and parameters of the poroelastic model.**

|  | BC | acGGM | AGX | acGX | acGGM + AGX | GGM$_{alk}$ + AGX |
|---|---|---|---|---|---|---|
| Compression–relaxation analysis |  |  |  |  |  |  |
| Thickness (mm) | 4.6 (0.1) | 4.3 (0.1) | 3.4 (0.2) | 4.8 (0.2) | 4.0 (0.2) | 4.7 (0.3) |
| Diameter (mm) | 41.9 (0.7) | 42.0 (0.1) | 42.6 (0.9) | 41.7 (0.4) | 42.3 (0.2) | 42.5 (0.2) |
| CR | 0.12 (0.03) | 0.15 (0.01) | 0.13 (0.02) | 0.15 (0.04) | 0.12 (0.01) | 0.13 (0.04) |
| $E_{relax}$ (kPa) | 1.7 (0.5) | 2.5 (0.2) | 0.8 (0.2) | 1.6 (0.2) | 1.3 (0.1) | 1.5 (0.2) |
| $G'$ (kPa) | 8.5 (0.3) | 13.0 (1.3) | 2.3 (0.7) | 12.6 (2.7) | 7.9 (2.1) | 9.1 (1.2) |
| $G''$ (kPa) | 1.1 (0.0) | 2.0 (0.2) | 0.3 (0.1) | 1.9 (0.5) | 1.1 (0.3) | 1.4 (0.1) |
| Parameters from the poroelastic model |  |  |  |  |  |  |
| $E_z$ (Pa) | 810 (101) | 1210 (29) | 332 (31) | 1049 (69) | 946 (63) | 923 (4) |
| $H_A$ (kPa) | 15.8 (1.8) | 32.1 (0.6) | 8.2 (3.9) | 22.6 (3.5) | 27.6 (2.6) | 23.0 (0.9) |
| $k \times 10^{-9}$ (m$^2$) | 12.2 (0.5) | 3.5 (0.1) | 10.9 (3.7) | 4.4 (0.5) | 3.2 (0.1) | 2.9 (0.2) |
| Uniaxial tensile testing |  |  |  |  |  |  |
| Dry content (wt%) | 8.4 (0.7) | 10.6 (1.0) | 8.2 (1.1) | 10.7 (0.6) | 11.7 (1.6) | 10.0 (0.6) |
| $\rho$ (mg cm$^{-3}$) | 82 (8) | 110 (6) | 90 (4) | 117 (28) | 118 (16) | 106 (8) |
| $E_{app}$ (MPa) | 2.4$^a$ (0.4) | 2.3$^a$ (0.3) | 0.4$^b$ (0.1) | 0.7$^c$ (0.1) | 2.4$^a$ (0.6) | 0.7$^c$ (0.2) |
| $\sigma_{max}$ (MPa) | 1.4$^a$ (0.2) | 0.9$^c$ (0.1) | 0.6$^b$ (0.1) | 0.9$^c$ (0.1) | 0.9$^c$ (0.2) | 0.5$^d$ (0.1) |
| $\varepsilon_{max}$ (%) | 53$^a$ (5) | 39$^b$ (2) | 87$^c$ (9) | 96$^e$ (4) | 41$^b$ (7) | 71$^d$ (10) |

Average values and SD (between brackets) from compression–relaxation mechanical analysis, mechanical and structural parameters from the poroelastic model, and uniaxial tensile testing. The sample names for the BC-H materials correspond to the hemicellulose content. Compression–relaxation mechanical analysis: $E_{relax}$, $G'$, and $G''$ values at a compression rate (CR) close to 0.1 and data for all CRs are presented in Supplementary Fig. 1. The data are presented as averages and SD from three to four replicates. Results from uniaxial tensile testing: SDs of dry content and density are based on three samples, and the stress–strain data on five to nine replicates. Data for pH 10 washed (deacetylated) BC-acGGM and BC pellicles are presented in Supplementary Table 3. Values with different superscripts (a–d) are significantly different at $p < 0.05$ (ANOVA single factor). Details of ANOVA analysis are in Supplementary Information (Supplementary Table 5).

behavior as the reference BC, potentially due to the effects of glucomannan and xylan hemicelluloses canceling one another out. These divergent effects demonstrate the distinct influence of the type and molecular structure of wood hemicelluloses on modulating the compression behavior of cellulose fibril networks.

**Modeling the poroelasticity of the hydrogels in compression.** Analysis of the micromechanics of BC-H hydrogel deformation under compression must take into account the biphasic nature of the porous network with the water phase of the hydrogel and the hydrated polymer network, monitoring the dynamics of water and the subsequent increase in density under compression[43,45]. The so-called poroelastic properties were analyzed by a modified version of the model described previously[45], based on the theory of stress relaxation under confined and unconfined compression of transversely isotropic materials[46]. The model describing a confined transversely anisotropic system showed the best fit to the data and was used to fit the poroelastic response of hydrogels during the relaxation phase (details of the model are provided in the Supplementary Methods). The parameters of the model (Fig. 3b) are the aggregate modulus ($H_A$), which is a function of the lateral (in-plane direction) modulus ($E_L$), the direct modulus in the (out-of-plane) direction of compression ($E_z$), and the hydrogel permeability ($k$)[44,47]. The model shows a consistently good fit for all BC-H hydrogels at a CR of 0.1 assuming that the permeability does not change as the hydrogels are compressed, which is verified in most parts of the compression cycles (Supplementary Fig. 9).

For all materials, increased CR resulted in higher $E_z$ and $H_A$ values, whereas for permeability the opposite trend was observed, with permeability values decreasing with increasing CR (Fig. 3d and Table 1). The observed trend for modeled $E_z$ and $H_A$ is similar to $E_{relax}$, whereby BC-acGGM and BC-AGX hydrogels stand out, having the highest and lowest moduli, respectively. Considering permeability, all hemicellulose containing hydrogels showed decreased permeability compared to the reference BC, indicating that wood hemicelluloses contribute to a more compact network, which restricts the water flow. A notable exception was the BC-AGX hydrogel, the permeability of which was comparable with that of pure BC and displayed a significant degree of variation as can be seen from the error bars shown in Fig. 3d. Previously, it was proposed that the water transport in poroelastic hydrogels follows a two-regime mechanism[45]. At short time scales, water drainage through the pores follows the hydrodynamic limit, whereby the flux (of mass) is proportional to the differential pressure. At longer time scales, the hydrated nature of hemicelluloses results in the diffusion-controlled water transport associated with water mobility in and out of the hydrated shell of hemicellulose polysaccharides. This hydrated hemicellulose shell with significant water interactions probably includes the "dangling" chains and the "loops" of adsorbed hemicelluloses in a "loops and trains" model, which contribute to the increase in the local viscosity of the fluid within pores, thus reducing the pore permeability. For BC-AGX composites, which show high permeability while being mechanically more compliant, it is possible to suggest that AGX adsorbs onto cellulose fibrils with fewer "dangling segments" and "loops,"

and, potentially, higher cellulose surface coverage. Such configuration would result in more open pores and at the same time it would reduce the formation of direct adhesive links between cellulose fibrils, thereby preserving the compliant nature of the BC network[48].

For all hydrogels, we observed a marked difference between the $E_{relax}$ and $G'$, which may be explained by considering material anisotropy. One of the possible ways of estimating the degree of anisotropy is through evaluating the effective anisotropy ratio $a$ (also known as the Zener ratio). The Zener ratio $a = 1$ corresponds to isotropic material, while for anisotropic materials $a > 1$. Here, the values of $a$ are found to be as high as 7–27 depending on the CR and the composition of the BC-H material. This indicates strong overall anisotropy for the BC-H hydrogels. The anisotropy of the lateral (in-plane) and the perpendicular (out-of-plane) dimensions of the hydrogels was further evaluated using the poroelastic model, involving the modeled aggregate modulus as described in the Supplementary Information (see Supplementary Eqs. 1–4 and Supplementary Table 2). The values of the Zener ratio in lateral in-plane dimension ($a_2$) are close to 1, which is consistent with transversely isotropic behavior of BC/BC-H. On the other hand, the anisotropy in the out-of-plane direction ($a_3$) is quite pronounced for all the BC-H hydrogels and increases upon the addition of hemicelluloses, suggesting that deformed fibers aligned in the out-of-plane direction contributes to the increased $G'$ upon compression. The overall results indicate that the aggregate modulus ($H_A$) and the shear modulus ($G'$) exhibit prominent correspondence and may describe the mechanical response of cellulose fibers during compression predominantly oriented in the horizontal direction of the BC and BC-H materials. These results reflect a layered type of structure of the BC-H hydrogels, such that compression from the top and bottom acts to squash the layers together, whereas in-plane compression would require some buckling or other deformations of component fibers. Polylamellate structures have been described in plant cell walls, both in primary cell walls after progressive enzymatic peeling[49] and in the layered structure of secondary cell walls[2], and this may be a general feature that is masked by non-cellulosic components. The orientation of cell wall lamellae is generally shown as being orthogonal to turgor pressure; therefore, the orientation in which BC-H hydrogels are compressed seems to be a reasonable analog for the compression effect of turgor pressure on plant cell walls.

**The extension of BC-H hydrogels is regulated by hemicellulose type.** Prior to tensile testing, the hydrated BC and BC-H pellicles were compressed to a solid concentration of about 10 wt% polysaccharide without drying, making the material more similar to the concentration in a hydrated plant cell wall prior to lignification. The mechanical response in tension was tested on these compressed materials, showing that the reference BC exhibits a maximum tensile strain ($\varepsilon_{max}$) of 53%, a maximum tensile stress ($\sigma_{max}$) of 1.4 MPa, and an apparent elastic modulus in the in-plane direction ($E_{app}$) of 2.4 MPa. The $E_{app}$ of pure BC was slightly lower compared to previous studies on the same type of BC (ATCC® 53524), although in highly hydrated state without compression[28,50]. The $\sigma_{max}$ values are similar (around 1.5 MPa) but the $\varepsilon_{max}$ values increased for the compressed samples in this work. This could be caused by the different fermentation times during the production of the BC hydrogels, as it has been shown that fermentation time has a strong influence on the mechanical properties in tension[50]. In our work, the fermentation time was considerably longer, possibly contributing to the relatively low $E_{app}$. A further interesting characteristic for the BC composites was the bimodal behavior at break, which has been previously attributed to the presence of cellulose chains with various lengths[50]. Considering the BC-H composites, the BC-acGGM showed a similar $E_{app}$ as pure BC but lower $\varepsilon_{max}$ and $\sigma_{max}$. The bimodal behavior upon break seems to be enhanced by the presence of wood mannans, which could be explained by the rigid interactions between mannan and cellulose surfaces promoting fiber slippage or pullouts. Interestingly, the xylan-containing hydrogels (BC-AGX and BC-acGX) both experienced a significantly lower $E_{app}$ at 0.4 and 0.7 MPa, respectively ($p < 0.05$), but a remarkable increase of $\varepsilon_{max}$ ranging from 87 to 96%, making the material less stiff and much more extensible. Nevertheless, only a modest effect on the mechanical properties in tension has been observed from incorporating cereal endosperm arabinoxylan and (1,3)(1,4)-β-glucan, respectively[28], whereas both pectin and xyloglucan cause a large increase in extensibility of related BC composite materials[25,42]. Here we observe that secondary cell wall xylans also contribute to a major increase of the tensile strain, which indicates the role of wood xylans in modulating the extensibility of secondary cell walls. The differences between the previously reported behavior for cereal arabinoxylan and our observations from wood xylans can be explained due to the molecular differences between both hemicelluloses. Arabinoxylan from cereal endosperm has a larger molar mass (in the range of 200–500 kDa), is highly substituted and does not have a patterned organization of the Ara substitutions along the xylan backbone. On the other hand, wood xylans (both spruce AGX and birch acGX) exhibit lower molar mass (in the order of 30 kDa) and show high content of GlcA and regular patterns with even-spaced substitutions along the xylan backbone. The patterned structure of wood xylans, together with the presence of uronic acid substitutions contributing to local charges in the hydrogel structure, seem to play a fundamental role in regulating the interactions with cellulose surfaces and the tendency to self-aggregate.

Surprisingly, the combined hemicellulose composites showed differences in their respective behavior, with the BC-acGGM + AGX composite having similar properties as BC-acGGM, whereas BC-GGM$_{alk}$ + AGX behaved more like BC-AGX. A plausible explanation for this behavior is the formation of multilayered structures surrounding the cellulose fibrils caused by the competitive differential interaction of the structural softwood hemicelluloses during BC biosynthesis and hydrogel assembly. Deacetylated GGM (GGM$_{alk}$) is known to form stronger interactions with cellulose compared to the acetylated counterpart (acGGM), which may have caused the migration of AGX to the more mobile regions of the BC-GGM$_{alk}$ + AGX hydrogels, thus promoting their ductility. In contrast, in the multicomponent BC-acGGM + AGX hydrogel, the combined interactions between AGX, acGGM, and the cellulose fibrils determine the network organization of the hydrogel and the mechanical properties.

The microstructure of the BC-H hydrogels after compression was visualized by SEM (Fig. 3e), showing that the BC microarchitecture becomes highly collapsed and dense after compression. The coalescence and association of fiber bundles can be clearly observed, which is induced by the adhesive behavior upon compression. Interestingly, the bundling of fibers into a collapsed structure is particularly significant for the mannan-containing hydrogels (BC-acGGM), whereas the presence of xylan seems to preserve the fibrillar architecture. Again, this is a consequence of the different interactions between hemicelluloses and BC, and the dispersive effects induced by the molecular patterning and the presence of uronic groups in xylan.

**Interactions of wood hemicelluloses with BC.** Solid-state $^{13}$C NMR analysis was used to investigate how the incorporation of

wood hemicelluloses influenced molecular conformations in the BC-materials. Cellulose crystallinity was estimated as 80% for the reference BC) system using peak integration of the cross-polarization (CP)/magic angle spinning (MAS) spectrum. The presence of spruce AGX reduced the crystallinity of the BC-H hydrogels to ~72% for the BC-AGX hydrogel and to 75% for the BC-acGGM + AGX, whereas acGGM seemed to increase slightly the overall crystallinity to around 83% for the BC-acGGM hydrogel (Supplementary Table 4). This indicates that the addition of xylan may hinder the packing of the cellulose bundles, whereas the presence of acGGM may help to facilitate cellulose crystallization. Interestingly, such decrease in crystallinity was not observed for the cereal arabinoxylan counterpart void of uronic acids, but has been reported for composites fabricated using tamarind xyloglucan and mixed-linkage β-glucan from barley[27,41,43]. In addition to this, the incorporation of well-defined wood hemicelluloses does not alter significantly the relative abundance of $I_\alpha$ and $I_\beta$ crystalline forms in the BC multicomponent hydrogels. A consistent transition from the more abundant $I_\alpha$ crystalline form in BC to the $I_\beta$ allomorph has been reported for tamarind xyloglucan[43], and for spruce mannan and beech xylan[30]. The implication is that the wood hemi-celluloses here studied are not able to interfere with cellulose microfibril assembly into mature fibers (as this would be expected to alter the $I_\alpha$ to $I_\beta$ ratio) and only interact with the surfaces of pre-formed fibers.

Solid-state NMR in the CP/MAS mode of the hydrated BC-H composites confirmed the occurrence of rigid interactions between cellulose and the purified wood hemicelluloses (Fig. 4a). GGM containing samples exhibited a peak at 102 p.p.m. (assigned to C1 in Man), which shows that GGM interacts closely with cellulose (Fig. 4b)[29]. The intensity of this peak increased for the BC hydrogel containing acGGM and subjected to a mild alkali deacetylation treatment (BC-deacGGM), which evidences that deacetylation enhances the interaction between cellulose and mannans. Considering both the birch (BC-acGX) and spruce xylan (BC-AGX) hydrogels, a less-intense xylan peak at C1 was detected compared to the mannan counterparts (Fig. 4b). However, the appearance of a peak at 82 p.p.m. corresponding with Xyl C4 (Fig. 4c) and the broadening of the peak at 63 p.p.m. attributed to C5 in xylan (Fig. 4d) indicates the occurrence of a rigid interaction with cellulose for the xylan containing BC hydrogels[14,51,52].

The fraction of each hemicellulose that experienced a rigid interaction with cellulose was estimated by comparing the C1 peak ratios between Glc and Man for acGGM, and the C4 peak ratios for Glc and Xyl for both acGX and AGX, taking into account the monosaccharide content determined for the respective BC-H pellicles. Based on this estimation, around 50–60% of incorporated hemicelluloses exhibited rigid interaction with cellulose, for both mannan and xylan populations in the BC-acGGM, BC-AGX, BC-acGX, and BC-acGGM + AGX samples. The estimated fraction of hemicelluloses experiencing rigid interactions increased significantly to around 100% for the mannan in the the BC-deacGGM pellicle subjected to a mild treatment at pH 10 for the removal of O-acetyl substitutions (Fig. 4b). The hemicellulose fraction experiencing rigid interactions with cellulose was also higher for the alkali-extracted (and thus deacetylated) GGM in the BC-GGMalk + AGX hydrogel compared to the native BC-acGGM + AGX hydrogel. This results indicate that adjusting the degree of acetylation ($DS_{ac}$) is of major importance for the interaction between hemicelluloses and cellulose in plant cell walls, as has been suggested by other authors in A. thaliana mutants with deregulated acetylation by knockout of a specific acetyltransferase[9] and by the deacetylation of glucomannans after cell wall formation[53]. Rigid interactions

were observed in this study for both birch acGX and spruce AGX, but they could not be detected in previous studies for cereal arabinoxylans and (1,3)(1,4)-β-glucans[28,43]. The occurrence of close interactions between xylans and cellulose seems to depend on the presence of regular motifs with even spacing of substitution, which is supported by previous work showing that a patterned xylan can adopt a twofold helical screw conformation that fits well onto cellulose surfaces[8,10,14,15]. Apart from the already mentioned rigid interaction between BC and seed mannans[29], this type of close interaction has also been detected for the primary cell wall hemicellulose xyloglucan[43]. As an illustration that not all hemicellulose is rigid in BC hydrogels, single-pulse MAS (SP/MAS) NMR spectra exhibit spectral patterns for wood hemicelluloses (Supplementary Fig. 10), consistent with their presence as more mobile species not rigidified by association with cellulose. The combined CP/MAS and SP/MAS NMR results suggest that the hemicellulose molecular structure modulates rigid interactions with cellulose surfaces for a significant fraction of the hemicellulose segments, whereas simultaneously other hemicellulose segments show flexible behavior modulating the bridging adhesion among the network components and the water permeability. This reinforces our hypothesis that both rigid and flexible segments in wood xylans and mannans are required to distinctly modify the mechanical properties of the cellulose hydrogels in tension and compression.

## Discussion

Hemicelluloses are a fundamental molecular component of lig-nocellulosic biomass contributing to approximately one-third of the total dry mass. However, their function in secondary cell walls and their contribution to biorefinery processes and bio-based products has received less attention compared to cellulose and lignin. In recent years, the integration of molecular plant biology, advanced characterization (using mass spectrometric glycomic analysis and solid-state NMR), and in silico molecular dynamic simulations has contributed to significant advances in under-standing the biosynthesis, molecular structure, and native con-formation in planta of secondary cell wall hemicelluloses. The occurrence of regular motifs with even patterning of glycosyl and acetyl substitutions has been reported for xylans from angiosperms[8,13,35,54,55] (hardwoods) and gymnosperms[10,15] (softwoods). This even substitution patterning in xylan plays a fundamental role in modulating the interactions with cellulose microfibril surfaces through a planar twofold conformation[9,10,14]. On the other hand, the presence of minor xylan domains with consecutive and clustered substitutions has been reported[10,13], which may have a fundamental role in the interactions with lignin[13,56]. Indeed, close steric contact and electrostatic interac-tions have been recently revealed between hemicellulose mobile domains and hydrophobic lignin aggregates in secondary cell walls from angiosperms (A. thaliana)[5] and also between mannan and xylan mobile domains and lignin in softwoods from conifers (spruce)[57]. The presence of regular substitution patterns in wood mannans remains elusive[36], although a recent study evidences that both xylans and mannans are intimately bound to cellulose surfaces in a semi-crystalline fashion in spruce wood[57]. It seems that an even placement of glucose and mannose units in the backbone of glucomannans benefits the interaction with cellulose surfaces[37,58], and preliminary evidence of the occurrence of such motifs in spruce glucomannan has been recently reported[37].

Despite these recent advances in understanding the molecular structure and supramolecular assembly of hemicelluloses in sec-ondary cell walls, the individual contributions of the different hemicelluloses to the mechanical properties of secondary cell

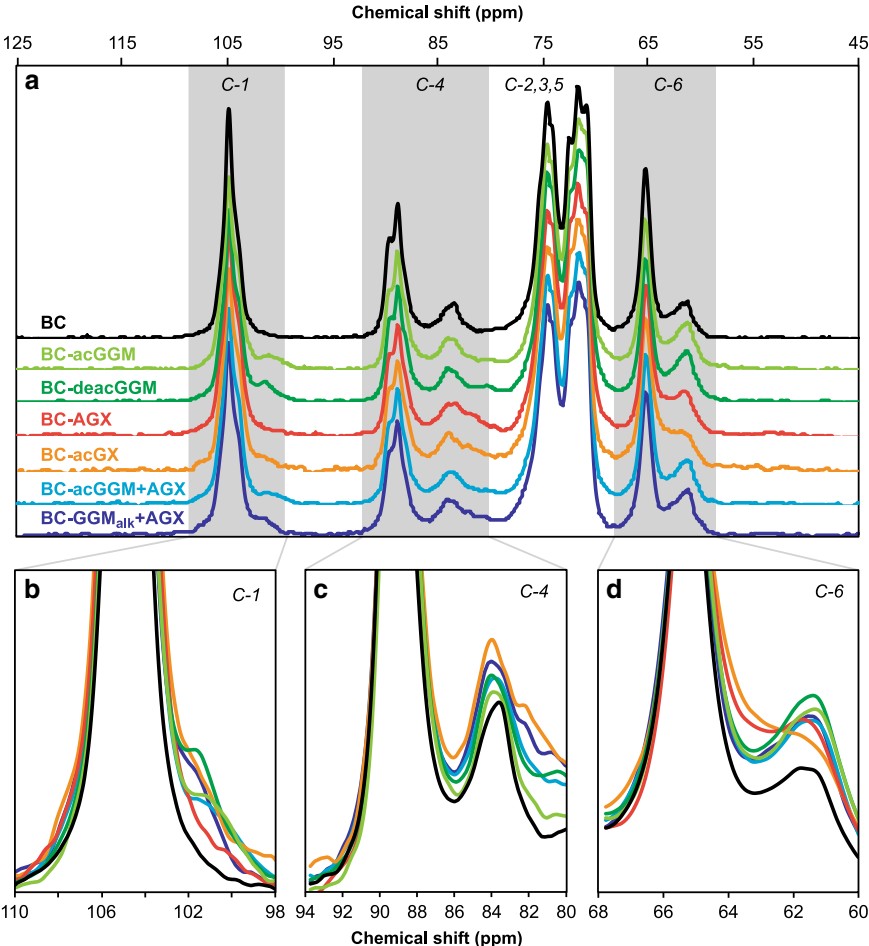

**Fig. 4 Interactions in BC-H hydrogels by solid-state $^{13}$C CP/MAS NMR analysis. a** Full spectra of the hydrated BC-H samples. **b–d** Magnification of the C-1 and C-4 regions of the NMR spectra for the spruce wood analogs, including BC-acGGM, BC-deacGGM (the same pellicle after a pH 10 wash for acetyl group removal), BC-GGM$_{alk}$ + AGX, and BC-acGGM + AGX. Source data are provided as a Source Data file.

walls and woody tissues are poorly understood. This remains a challenge due to the lack of sensitive and selective methods to probe mechanical properties in the distinct nanodomains of the plant cell wall. Therefore, relevant models that would allow a systematic investigation of the contribution of different hemicelluloses and molecular motifs to the overall mechanical performance are needed. Here we have used a BC system that produces cellulose fibrils extracellularly in the presence of purified and well-defined wood hemicelluloses extracted from spruce and birch as model systems of softwood and hardwood cell walls prior to lignification. This results in the spontaneous self-assembly of cellulose–hemicellulose composites under quiescent and highly hydrated conditions. Multicomponent BC–H hydrogels with controlled compositions were fabricated in large amounts, which enabled the systematic study of the biomechanical properties in compression and tension, their morphology, and the molecular interactions between the hemicelluloses and cellulose bundles. The targeted and homogeneous incorporation of the hemicelluloses was demonstrated by the compositional analysis of the hydrogels and the localization with selective antibodies. The presence of hemicelluloses did not significantly alter the morphology of the cellulose bundles (as observed by SEM) or their crystallinity (as revealed by solid-state NMR), which suggests that molecular interactions were confined to the surfaces of cellulose bundles rather than interfering with the assembly of the microfibrils at the stage of being produced by cellulose synthase action, although hemicellulose incorporation between bundles cannot be

discarded[30]. However, significant differences in the biomechanical responses were observed for the different BC-H hydrogels depending on the type of hemicellulose. Acetylated GGM (acGGM) from spruce mainly contributed to an increased modulus in compression and this effect was even more pronounced when the acetyl groups were removed by an alkaline treatment (GGM$_{alk}$). On the other hand, arabinoglucuronoxylan (AGX) from spruce and acetylated glucuronoxylan (acGX) from birch, both improved remarkably the extensibility of their compressed hydrogels during tension. Wood mannans influence the elastic modulus in compression, whereas wood xylans dramatically increase the ductility of BC hydrogels. This distinct behavior of the biomechanical responses under compression and tension can be directly observed by the comparison of the E-moduli from compression and tension in Fig. 3f. In general, the elastic modulus under tension is an order of magnitude higher for all the BC-H hydrogels compared to the compression values. The modulus from the compression–relaxation experiments ($E_{relax}$ at a CR ≈ 0.1) represents the viscoelastic response of the highly hydrated network. On the other hand, the modulus in tension was obtained after the compression of the hydrogels, which resulted in a less hydrated network and increased contact between cellulose bundles. Therefore, the influence of the hemicellulose phases on the bundle–bundle contacts during extension are mainly evaluated during the tensile testing, whereas the poroelastic properties of the hydrated network in compression offers information about the potential adhesion between the hemicellulose matrix and the

BC fibers. This comparison suggests that wood xylans strongly influence bundle–bundle contact resulting in a remarkable plasticizing effect during extension, whereas spruce GGMs seem to contribute to the aggregation and toughening of cellulose bundles, which result in an increased modulus during compression. The viscoelastic properties of the BC-H hydrogels under compression were evaluated using a poroelastic biphasic model, which considers the flow mechanics of water as an incompressible fluid through an elastic solid soft material phase (in our case, the BC-H hydrogels). The modeling of the experimental data to the poroelastic theory demonstrated the high anisotropy of the BC-H hydrogels, especially in the in-plane lateral dimension perpendicular to the direction of compression. The addition of hemicelluloses generally contributed to this lateral anisotropy and to a decreased permeability of the BC-H hydrogels under compression. In addition to this, the contribution of the modulus in the out-of-plane direction of compression proved to be negligible compared to that of the aggregate modulus, which strongly depends on the lateral mechanical resistance of the material. This indicates that the mechanical response during compression can be predominantly explained by the oriented bundles in the horizontal direction of the BC-H materials.

The distinct biomechanical contribution from wood hemicelluloses can be explained from the microstructure of the BC-H hydrogels and the nature of the interactions with cellulose surfaces, their hydration, and their aggregative effects, which in turn are modulated by the presence of decoration motifs in their molecular structure. All the studied wood hemicelluloses exhibit rigid interactions with cellulose when incorporated in the BC-H hydrogels, as revealed from solid-state NMR, demonstrating that both glucomannans and xylans can bind tightly to cellulose. However, no clear differences can be observed in the relative abundance of rigid interactions of spruce (AGX) and birch (acGX) xylans with cellulose surfaces in the BC-H hydrogels, as ~50% of each incorporated hemicellulose is immobilized, most likely because it is tightly bound with cellulose, with the other 50% more flexible and likely to be located in the gaps between cellulose bundles. In vitro analyses of the adsorption of hemicelluloses onto different cellulose model surfaces by Quartz Crystal Microbalance with Dissipation monitoring have also indicated, in most cases, the occurrence of two layers of adsorbed hemicellulose polymers with distinct viscoelastic behavior[59–61], which seems to be influenced by the specific solubility and aggregative properties of the hemicelluloses[62]. The presence of both phases (rigid and flexible) seems to be required to explain the increase of modulus during compression and ductility during extension. The differences in biomechanical behavior between xylans and mannans cannot be then explained solely from the nature of their interactions with cellulose, but from the multiscale architecture and the aggregative effects among hemicellulose-coated fibrils. The incorporated wood xylans (acGX and AGX) have a regular presence of uronic acid substitutions that might on one hand coat the surface of the cellulose bundles but at the same time provide local electrostatic repulsions among fibrils and bundles. Indeed, higher values of zeta potential have been reported for spruce AGX (−20 mV)[38] compared to spruce acGGM (in the order of −10 mV[39]). On the other hand, the presence of regular decoration motifs in AGX and acGX allow a tight interaction with cellulose surfaces through a twofold conformation, which may coat the surface of the cellulose microfibrils through a rigid layer (Fig. 5a). At the same time, the presence of non-patterned minor domains with uneven and consecutive glucuronation might prevent extensive binding and favor a more mobile conformation in the flexible phase. This indicates that xylan coating of cellulose fibers and bundles is "non-sticking" showing weak bridging adhesion, and therefore

acts as a dispersant agent preventing the coalescence of cellulose bundles in compression and lubricating the cellulose–cellulose contact points in tension resulting in a much easier (low modulus) and greater (increased strain to break) fiber pull-out. This is in agreement with the occurrence of both steric and electrostatic forces (so-called electrosteric effects) between the flexible xylan–xylan domains coating the BC bundles, as previously reported in in-vitro systems[63,64]. Other non-woody xylans previously used in BC hydrogels (e.g., arabinoxylan originating from the wheat endosperm) do not contain glucuronic acid substitutions or regular domains. These hydrogels did not show the presence of a rigid phase, and did not have the extension enhancement here presented. Therefore, the controlled molecular structure of the wood xylans ensures the organization of the polysaccharide networks into both rigid and flexible phases, which in turn influences the mechanical properties of the hydrogels.

On the other hand, glucomannans are decorated with acetyl groups and galactose side chains, they do not contain acidic substitutions, and have a stiffer backbone configuration than xylan[65]. Native acGGM is capable of showing rigid interactions with cellulose as demonstrated by CP/MAS NMR and therefore can coat cellulose fibers and bundles; the occurrence of such rigid interactions is enhanced by chemical deacetylation. Moreover, the flexible and hydrated mannan segments are capable of creating aggregated layers that can bridge adjacent cellulose bundles when compressed in a way that is not possible for pure BC hydrogels, therefore acting as a gluing agent between cellulose fibrils and bundles (Fig. 5b). This indicates that the mannan-coated bundles exhibit strong bridging adhesion, which can be attributed to the stickiness and the aggregative nature of the flexible mannan domains, and therefore increases the mechanical properties under compression and does not assist in sliding the cellulose bundles under tension as proposed for the wood xylans, and therefore does not affect extensibility. This interpretation is supported by the permeability results from the poroelastic model and is in agreement with the observations on wood kraft pulps where glucomannan resulted in improved fiber strength[66] and the closer cooperation, higher affinity, and tighter association between glucomannan and cellulose[27] compared to that between xylan and cellulose[32,60,67].

Interestingly, different effects were observed when acGGM and AGX were incorporated separately into a cellulose microfibril network as compared to when they were both combined. This indicates that adjusting the content and type of hemicellulose might be a way to control the mechanical properties of secondary cell walls. Indeed, spruce wood contains around 20% acGGM and 10% AGX[68–70] (a hemicellulose ratio reflected also in the combined composites), whereas birch contains around 27% acGX and only around 3% glucomannan[13,22,69,70]. The birch composite in this work, BC-acGX, contained mainly acGX but also low amounts of co-extracted glucomannan and in compression, the BC-acGX behaved in a similar way as the combined spruce composites (BC-acGGM + AGX and BC-GGM$_{alk}$ + AGX). This again suggests that the biomechanical role of different wood hemicelluloses differs and that the varying ratios of glucomannan and xylan hemicelluloses in hardwood and softwood are modulated during biosynthesis depending on their structures contributing to the architecture and properties of the cell wall.

Although we are aware that the BC system has structural and morphological differences with the plant model, we believe that our BC–wood hemicellulose hydrogels offer important insights into the likely biomechanical role of hemicelluloses in secondary plant cell walls and wood materials. The differences between the cellulose originating from wood and produced by the bacterium in our model system include the individual microfibril geometry

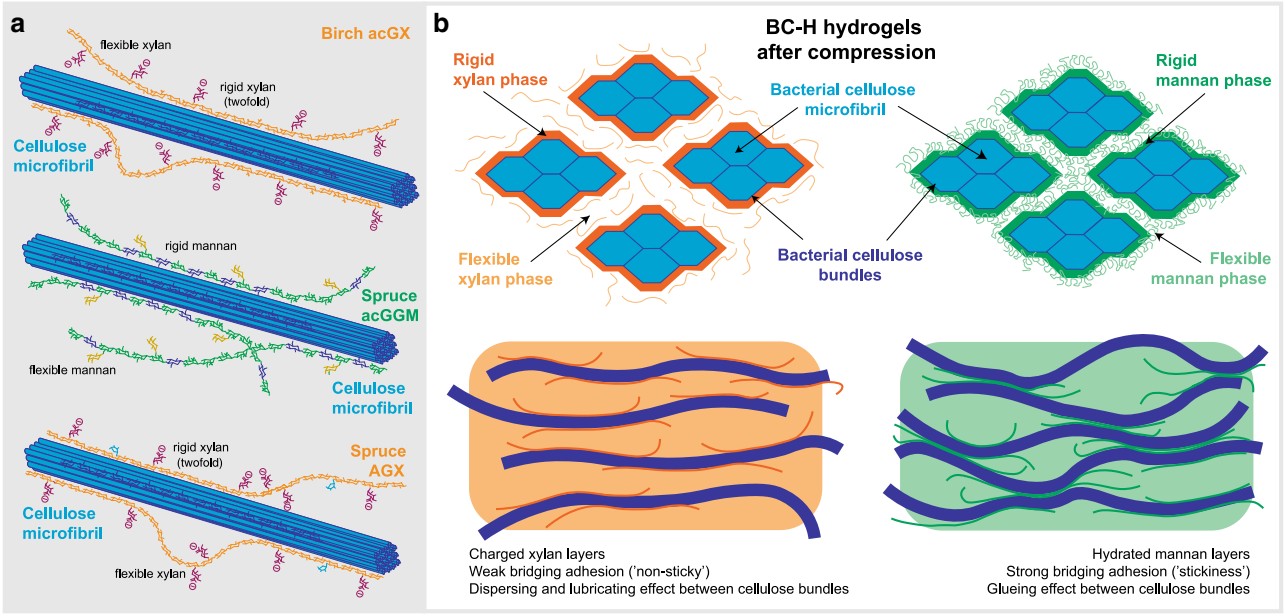

**Fig. 5 Proposed organization of the BC-H hydrogels with incorporated wood hemicelluloses. a** Molecular interactions between the cellulose microfibrils and the wood hemicelluloses: birch xylan (acGX), spruce glucomannan (acGGM), and spruce xylan (AGX). Here we depict individual xylan and glucomannan polymers that exhibit both regular molecular motifs interacting with the cellulose surfaces in a rigid phase and non-patterned domains in a flexible phase. **b** Proposed scheme for the architecture of the BC-H fibrillar networks. In the rigid phases, the hemicelluloses attain extended conformations directly interacting with the cellulose surfaces. In the flexible phases, the hemicelluloses may adopt more coiled conformations where they can interact with each other through bridging adhesion of different intensities (stronger for glucomannans, weaker for xylans).The presence of rigid and flexible phases in BC-H hydrogels has been estimated as 50% for each phase, based on the results from $^{13}$C CP/MAS NMR analysis.

and dimensions, the relative abundance of the $I_\alpha/I_\beta$ crystalline allomorphs[23], the orientation of the cellulose bundles[71], as well as the presence of lignin affecting the wood cell wall properties. These factors should be considered when discussing the properties of the observed hydrogels in relation to plant cell walls. Lignin and lignin–carbohydrate complexes are expected to limit elastic deformation of the lignified cell walls[72]. Lignin has previously been considered in similar systems where BC containing lignin–carbohydrate complexes (covalently connected lignin and hemicellulose moieties) were fabricated. The lignin moieties did not show any direct interaction with the cellulose hydrogels, but were suggested to be essential for the cellulose–hemicellulose–lignin framework[32]. This is in agreement with the latest models for secondary cell wall organization, where lignin is depicted as having no direct contacts with cellulose surfaces, but extensive electrostatic interactions with flexible xylan and mannan domains[5,57]. This architecture is well reflected in our BC-H hydrogels, suggesting that hemicelluloses mediate the supramolecular connectivity between cellulose surfaces and lignin and the overall cell wall integrity. Indeed, our model integrates the current knowledge on hemicellulose interactions with cellulose surfaces[9,14,57] and the complex solution/aggregation properties of hemicelluloses[38,39,61]. Computational modeling by molecular dynamic simulations has recently contributed to a fundamental understanding of the driving forces for hemicellulose – cellulose interactions[8,10,61,73]. However, these studies are currently limited to few molecular motifs and small scales, as the simulation of the multiscale interactions between the cellulose microfibrils and the rigid/flexible hemicellulose phases would require vast computational efforts. Our results constitute the starting point for the future development of multiscale coarsed grain modeling of plant cell wall assembly. Moreover, our study reveals for the first time the distinct biomechanical contributions of wood xylans and mannans to the biomechanical properties of cellulose hydrogels mimicking softwood and hardwoods prior to

lignification. Our multiscale model proposes that the rigid and flexible interactions between cellulose and hemicelluloses are responsible for the distinct compression and tension behavior, with fundamental implications for the function of secondary plant cell walls. In addition, the results have significance for the preparation of lignocellulosic-based materials with targeted properties, where hemicellulose molecular structure and content can be used to induce orientation of cellulose nanofibres[74] and to alter the ductility of cellulose-derived membranes and hydrogels[75,76]. Indeed, the molecular structure of wood hemicelluloses seems to largely influence the assembly of cellulose microfibrils into higher-ordered structures through both rigid and flexible domains with plasticizing and antiplasticizing properties, which in turn may assist in controlling and tuning the biomechanical properties of the derived materials in tension and compression.

## Methods

**Wood materials**. Thermomechanical spruce pulp (TMP) was used for acGGM extraction, whereas wood chips were used as starting material for alkaline extraction of spruce and the subcritical water extraction from birch. The wood chips were processed by a Wiley mill (40 mesh cutoff) followed by acetone extraction according to SCAN-CM 49 : 03 to remove low molecular weight extractives.

**Extraction of wood hemicelluloses**. Spruce AGX and GGM were extracted by alkaline extraction[16,77] from spruce TMP after chlorite delignification[77–79]. The acetone extracted spruce wood powder and deionized $H_2O$ were mixed by a stirrer at a ratio of 1 : 15 and heated at 75 °C. Once every hour, 0.1 ml glacial acetic acid per gram wood and 0.3 g $NaClO_2$ per gram wood were added for a total of seven times. After a total treatment time of 8 h, the delignified wood powder (holocellulose) was filtered, washed with excess of deionized $H_2O$, and dried. The alkaline extraction of hemicelluloses was performed at room temperature with a 1 : 10 ratio of holocellulose : KOH (24%) for 24 h at constant stirring and under $N_2$ atmosphere. The liquid was collected by filtration and extracted hemicelluloses were precipitated at a ratio of 1 : 4 : 0.4 filtrate : ethanol : acetic acid followed by centrifugation, washing in 75% ethanol, and freeze drying. Part of the extracted hemicelluloses were further processed to separate the AGX and GGM fractions by

Ba(OH)$_2$ precipitation of GGM[77]. The extracted hemicellulose was dissolved in 10 % KOH and a solution of 5% Ba(OH)$_2$ was added dropwise at a ratio of 15 g hemicellulose : 200 ml KOH : 400 ml Ba(OH)$_2$. The precipitated GGM fraction was separated from the AGX rich solution by centrifugation, and the GGM was recovered by dissolution in 5% KOH, acidification by HAc to pH 3, and precipitation by EtOH. AGX was precipitated from the solution in the same ratios as previously described. Both the GGM and AGX fractions were centrifuged, washed by EtOH, dissolved in Milli-Q® (MQ) H$_2$O, and freeze dried.

Acetylated GGM (acGGM) was extracted from spruce TMP by warm water (60 °C) followed by EtOH precipitation[33]. In brief, the spruce TMP was mixed with deionized water under stirring for 3 h at 60 °C. The supernatant was collected by filtration and the extracted acGGM was precipitated at a ratio of 1 : 4 filtrate : ethanol, followed by subsequent centrifugation, resuspension in deionized H$_2$O, and freeze drying.

Birch acGX was extracted from acetone-washed birch wood powder by subcritical water extraction in an accelerated solvent extractor using sodium formate buffer (pH 5, 0.2 M)[13]. Each cell was loaded with 5 g wood powder and four consecutive extractions for 5, 15, 20, and 20 min at 170 °C were made on the same cell and collected in separate glass vials. The fraction from the first 5 minutes was discarded due to the high mannan content, and the subsequent three fractions were combined and pooled together. The solution was dialyzed at 6–8 kDa cutoff and freeze dried. The extracted birch xylan was further purified by EtOH washing precipitation to remove residual lignin before a final freeze drying.

**Characterization of wood hemicelluloses.** The molecular weight of the polysaccharides was analyzed on a SECurity 1260 system (Polymer Standard Services, Mainz, Germany) equipped with refractive index (RI), ultraviolet, and multiangle laser light scattering (MALLS, BICMwA7000, Brookhaven instrument Corp., USA) detectors. The mobile phase consisted of 0.1 M NaNO$_3$ and 5 mM NaN$_3$ at a flow of 0.5 ml min$^{-1}$. Separation was performed by a PSS Suprema guard column (50 × 8 mm, 10 µm particle size), a 30 Å, and two 100 Å PSS Suprema columns (300 × 8 mm, 10 µm particle size) connected in series at a temperature of 40 °C. An RI increment (dn/dc) of 0.146 mL g$^{-1}$ for pullulan was used for MALLS estimation[80] and ten pullulan standards ranging between 342 and 708,000 Da were used for standard calibration.

The monosaccharide composition of the extracted wood hemicelluloses was analyzed by acid methanolysis[81,82] and anion exchange chromatography with pulsed amperometric detection (HPAEC-PAD, ICS-3000, Dionex, Sunnyvale, CA, USA). The freeze-dried samples were incubated with 2 M HCl in dry methanol at 1 mg/ml for 5 h at 100 °C. The samples were dried under N$_2$ gas and a second hydrolysis step in 2 M trifluoroacetic acid (TFA) at 120 °C for 1 h was performed. The samples were dried again and dissolved in MQ-H$_2$O. The hydrolyzed monosaccharides were analyzed by HPAEC-PAD at a flow rate of 1 mL min$^{-1}$ equipped with a Dionex CarboPack PA1 column at 30 °C. Two gradients (solvent A: deionized water; solvent B: 300 mM sodium hydroxide; solvent C: 200 mM sodium hydroxide + 170 mM sodium acetate; solvent D: 1 M sodium acetate) were used for the quantification of neutral sugars and uronic acids, respectively[83]. Neutral monosaccharides were separated in 100% (v/v) solvent A after equilibration of the column for 7 min with 60% (v/v) solvent B and 40% (v/v) solvent C, ramped to 100% (v/v) solvent A for 1 min, and further equilibrated at 100% (v/v) solvent A for 6 min prior to injection. Detection of the neutral monosaccharides required post-column addition of 0.5 mL min$^{-1}$ of 300 mM sodium hydroxide. The uronic acids were analyzed with a 30 min gradient from 10% (v/v) solvent B to 10% (v/v) solvent B + 40% (v/v) solvent D.

Calibration curves were made with Rha, Fuc, Man, Glc, Xyl, Ara, Gal, GlcA, GalA, and 4-O-mGlcA. Samples were hydrolyzed in triplicates.

Solution-state $^1$H NMR was used to study the chemical structure comparing the monosaccharide ratios of the extracted hemicelluloses with the hemicellulose washed out at harvest to determine if some structures showed a higher tendency to incorporate. The extracted hemicelluloses and the precipitated and freeze-dried samples from the first wash and the growth medium were dissolved for NMR analysis using sodium 3-(trimethylsilyl) propionate-2, 2, 3, 3-d$_4$ as an internal standard. $^1$H NMR spectra were measured on a Bruker Avance 700 MHz spectrometer (Bruker, Billerica, MA, USA) operating at 343 K equipped with a 5 mm CPTCI probe. $^1$H NMR spectra were recorded using excitation sculpting Bruker pulse sequence "zgesp" was used with a 11 s relaxation delay and 32 scans.

The DS$_{ac}$ was determined by saponification followed by high-performance liquid chromatography (HPLC). About 7 mg of sample was mixed with 300 µl MQ-H$_2$O, 1.2 ml 0.8 M NaOH, and 10 µl of 1 M propionic acid (as internal standard) in a thermomixer at 60 °C overnight. The solution was neutralized with 37% HCl and the concentration of acetic acid was determined by analysis on an Ultimate-3000 HPLC system (Dionex Sunnyvale, CA, USA) equipped with a Phenomenex Rezex ROA organic acid column together with a standard curve of known acetic acid concentrations[84]. The samples were analyzed in duplicates and the DS$_{ac}$ was calculated according to Eq. 4:

$$DS_{ac} = \frac{MW \times \%acetyl}{\left(M_{acetyl} \times 100\right) - \left(M_{acetyl} - 1\right) \times \%acetyl} \quad (4)$$

Here, MW is 132 g mol$^{-1}$ for xylan and 162 g mol$^{-1}$ for mannan (molecular weight for a dehydrated xylose and mannose unit, respectively), $M_{acetyl}$ is the

molecular weight of acetyl groups (43 g mol$^{-1}$), and %acetyl is the concentration of acetic acid determined by HPLC. The DS$_{ac}$ was adjusted for the main hemicelluloses taking into account the exact composition from the monosaccharide analysis.

The distribution between C$_2$ and C$_3$ acetylation on extracted hemicelluloses were estimated by 2D heteronuclear single quantum coherence NMR (HSQC NMR) spectroscopy. Approximately 10 mg of sample was dissolved in 700 µl D$_2$O and analyzed by the HSQC method for 85 scans[85]. The C$_2$ and C$_3$ distribution were estimated form integrating the respective peaks[13,85].

The oligomeric mass profiling of the wood hemicellulose fractions was analyzed by enzymatic digestion and electrospray ionization mass spectrometry. In brief, 1 g L$^{-1}$ solutions of the xylan and mannan fractions were hydrolyzed using a β-glucuronoxylanase from GH30[34] (gift from Professor James Preston, University of Florida) and a β-mannanase from family GH5[86] (Nzytech, Portugal) for 16 h at 37 °C at pH 6 in acetate buffer (50 mM). After incubation, the endpoint digestions were boiled for 10 min, filtered with Amicon centrifugal speed-filters of 10 kDa (Merck Milipore, USA), desalted with HyperSep™ Hypercarb™ cartridges (Thermo Fisher, UK) and redissolved in 50% acetonitrile, 0.1% formic prior to analysis. The samples were directly infused at a rate of 0.1 ml min$^{-1}$ into a Synapt G2 (Waters, USA) mass spectrometer in positive mode, with a capillary and cone voltage of 2.2 kV and 60 V, respectively.

**Preparation of BC-H hydrogels.** The BC-H hydrogels and reference BC pellicles were produced using HS medium[87], where all the water was replaced by coconut water (CC-H$_2$O, Pure Natural, Raw C, Australia)[88]. Hemicellulose was dissolved at a concentration of 1% in CC-H$_2$O by magnet stirring at 60 °C overnight. Equal parts of hemicellulose solution and double concentrated CC-H$_2$O-based HS medium (pH 5, 2 % Glc) were combined resulting in the hemicellulose final concentration being 0.5%. One loop of ATCC® 53524 colonies, a K. xylinus strain, was added to the medium (20 ml/loop), and the primary inoculum was incubated at 30 °C for 72 h in a round yellow lid container (4 cm diameter, 70 ml total volume capacity). Fresh hemicellulose solution was prepared for scale-up in the same ratios as described above, and 18 ml was added to a separate yellow lid container followed by 2 ml of the primary inoculum. The containers were incubated at 30 °C for 7 days (168 h), and for AGX an additional batch that was incubated for 10 days (240 h) due to lower thickness than other BC pellicles after 7 days. For each BC-H composite preparation, at least 16 hydrogels were grown simultaneously as replicates. The pellicles were harvested by washing in cold sterilized MQ-H$_2$O under gentle magnet stirring. Each wash lasted for at least 15 min and nine changes of water were made. The pellicles were thereafter stored in 0.02% NaN$_3$ at 4 °C. To investigate whether specific hemicellulose structures tend to stay in solution instead of incorporating with the cellulose, the first wash of some pellicles was collected separately. The polysaccharides remaining in the liquid were precipitated by ethanol and the solid fraction was separated by centrifugation, dissolved in MQ-H$_2$O, and freeze dried. Also, the remaining growth medium was precipitated, centrifuged, and dried in the same way.

A mild alkaline treatment was developed to deacetylate pure acGGM and then applied on the BC-H pellicles. In brief, a ratio of 1 pellicle to 100 ml buffer solution (pH 10.2, 67 mM NaHCO$_3$ and 37 mM NaOH) was gently stirred for 24 h at room temperature. The pellicles were then washed with sterile MQ-H$_2$O and finally stored in 0.02 % NaN$_3$ at 4 °C.

**Structural characterization of BC-H hydrogels.** The monosaccharide composition of the BC-H hydrogels was determined by sulfuric acid hydrolysis[89], to establish the incorporation level of hemicelluloses into composites. The BC-H pellicles were freeze dried and a sample of 2 mg was treated with 125 µl of 72% H$_2$SO$_4$ at room temperature by intermittent vortex mixing for 3 h. Thereafter, 1375 µl MQ-H$_2$O was added and the hydrolysis continued at 100 °C for another 3 h. The samples were allowed to cool, diluted 1 : 10 in MQ-H$_2$O, and analyzed in triplicates by HPAEC-PAD as presented before.

The distribution of the hemicelluloses within the BC-H hydrogels was assessed by antibody labeling and confocal microscopy[90]. Small pieces (<2 × 2 mm) of the pellicles were prepared by a scalpel and placed in excess of blocking buffer (1% bovine serum albumin in phosphate buffered saline (PBS), pH 7.2) for 30 min. The supernatant was replaced by primary antibody solution (1/20 dilution in blocking buffer) and the samples were incubated at room temperature for 2 h. LM11 (anti-xylan) and LM21 (anti-mannan) were applied separately (Plantprobes, UK). The supernatant was removed and the samples were washed by the addition of PBS, which was left for at least 5 min and then replaced with fresh solution. This was performed three times with PBS and three times with blocking buffer. Diluted secondary antibody Alexa Fluor® 555 (Thermo Fisher) (1/50 in blocking buffer) was added and the samples were incubated for 2 h in a dark place followed by washing in the same way as before. The sample was placed on a microscope slide; a drop of Vectashield (Vector Laboratories, Inc., Burlingame, USA) was added followed by a cover slide. The analysis was performed on a LSM700 confocal microscope (Carl Zeiss, Germany) at 555 nm wavelength, and the emission spectrum was automatically adjusted by the Zen 2011 software coupled with the microscope.

The morphology of the BC and BC-H hydrogels was evaluated by SEM. The hydrogels were cut into 4x4 mm pieces with a scalpel, snap-frozen in liquid

nitrogen for 10 s and thereafter directly transferred to ice-cold 3% glutaraldehyde in methanol. The samples were stored at −20 °C for 24 h, transferred to pure MeOH and stored another 24 h at −20 °C, and finally moved to absolute ethanol at room temperature until critical point drying and SEM analysis. The samples were analyzed on a scanning electron microscope (XL30, Philips, Eindhoven, Netherlands) under an accelerating voltage of 5 kV, both before and after compression.

The architecture structure of the BC and composites were examined using solid-state $^{13}C$ CP/MAS and SP/MAS NMR spectroscopy experiments at a $^{13}C$ frequency of 75.46 MHz on a Bruker MSL-300 spectrometer (Bruker, Billerica, MA, USA). Excess liquid water was squeezed from the pellicles and they were packed in a 4 mm diameter, cylindrical, PSZ (partially stabilized zirconium oxide) rotor with a KelF end cap. All CP/MAS and all SP/MAS spectra were acquired on hydrated samples after gently squeezing out excess water. The rotor was spun at 5 kHz at the magic angle (54.7°). The 90° pulse width was 5 µs and a contact time of 1 ms was used for all samples with a recycle delay of 3 s. The spectral width was 38 kHz, acquisition time 50 ms, time domain points 2 k, transform size 4 k, and line broadening 50 Hz. At least 5000 scans were accumulated for each spectrum. Spectra were referenced to external adamantane. The percentage of bound hemicellulose in each composite sample was determined by calculating the relative areas of the peaks at 105 p.p.m. (Glc C1) and 102.5 p.p.m. (Man C1) for mannans, and the areas at 89 and 84 p.p.m. (ordered and amorphous cellulose C4, respectively), as well as 82 p.p.m. (Xyl C4) for xylans. The crystallinity was estimated by integration of the crystalline cellulose region at 88–92 p.p.m. and the non-crystalline region at 83–86 p.p.m. for C4 in Glc[91]. The crystalline region was analyzed to determine the ratio between Iα and Iβ cellulose[28].

**Mechanical testing of BC-H composites.** A HAAKE Mars III Rheometer (Thermo Fisher Scientific, Karlsruhe, Germany) was used to analyze the mechanical properties of the hydrogel pellicles in compression using small amplitude oscillatory deformation analysis[45]. The rotational rheometer was equipped with 6 cm parallel titanium plates covered with sandpaper (P240, 58 µm roughness) to prevent sample slip. The bottom dish of the measuring geometry made it possible to perform the tests in solution where water is free to flow in the radial direction of the pellicle. Before each measurement, the gap was zeroed at a normal force of 4 N avoiding errors due to the presence of air. The BC pellicle was placed at the center of the bottom plate, MQ-$H_2O$ was added, the upper plate was lowered closer to the pellicle without touching, and additional MQ-$H_2O$ was added to cover both the sample and the upper plate. The measurements were performed at 25 °C regulated by a Peltier element and axial compression was performed at a rate of 5 µm s$^{-1}$. Initially, the upper plate was lowered until a normal force ($F_n$) of 0.3 N was recorded; then the plate was raised up and lowered again to a $F_n$ of 0.5 N. This position is considered as the starting thickness ($h_0^*$) of the pellicle. This pre-compression was performed to assure a reproducible starting point where the gel contact with the plate is maximized and can be approximated by the nominal geometric area of the BC disks[92]. The gap was increased again to equilibrate the sample under the condition of zero load, and then decreased again at a constant rate (5 µm s$^{-1}$) until a specific force was reached. The sample was allowed to relax under the constant gap condition. The $F_n$ values were recorded as a function of time and gap for each compression–relaxation cycle. After the relaxation step was completed, oscillations were performed to evaluate the storage ($G'$) and loss moduli ($G''$). The frequency of 1 Hz was chosen based on previous data[45,92] and the stress range was 2–4 Pa in the linear viscoelastic regime as determined by a sweep between 0.1 and 5 Pa (Supplementary Fig. 6). Thereafter, the upper plate was increased until it was no longer in contact with the sample; this step was introduced to quantify the degree of irreversible deformation of the sample caused by the applied force. The same procedure was repeated at $F_n$ set points of 1, 1.6, 2.5, 4, 6.3, 10, and 30 N. The $G'$ and $G''$ values were adjusted to account for the diameter of the sample. Each hydrogel was analyzed in triplicates and averages are presented. The methodology is referred to as Method 1 in Supplementary Table 2. To probe anisotropic effects in more detail, a second methodology (Method 2) was used on a selected set of samples (duplicates of BC, acGGM, AGX, and acGX). The procedure was the same as in Method 1 except that only four normal force points set at 1.6, 4, 10, and 30 N were used. Another difference was that after the oscillation sweep at each set point the gap was slowly increased to reach the position where $F_n$ is ~0 N but both plates are still in contact with the pellicle. Here, a second oscillation sweep was performed before proceeding to the next force set point cycle.

For the uniaxial tensile testing, the pellicles were compressed using a HAAKE Mars III Rheometer (Thermo Fisher Scientific, Karlsruhe, Germany) under constant $F_n$ of 30 N. Thereafter, the pellicles were sandwiched between two glass slides and were put under constant weight of 1.54 kg (~15 N) for 24 h in a water bath (MQ-$H_2O$) at room temperature. After compression, the pellicle-glass slide assembly was taped together using an adhesive tape and stored in 0.02% NaN$_3$ at 4 °C until tensile testing. The sample specimens for tensile testing were prepared by a dumbbell cutter (ISO 37-4, DILCO, Ontario, Canada) and the sample thickness was measured by a bench-top micrometer using an average of 3 measurements on the same pellicle The uniaxial tensile testing was performed using an Instron 5543 (Instron, Melbourne, Australia) on a 5 N load cell at a constant speed of 10 mm min$^{-1}$ [50]. The force-deformation data were converted to stress–strain curves and the apparent Young's modulus ($E_{app}$) was determined in the linear region at either

5–25% strain (for BC, BC-acGGM, and BC-acGGM + AGX composites) or 10–30% strain (BC-AGX, BC-acGX, and BC-GGM$_{alk}$ + AGX composites). The diameter (measured by a digital caliper) and weight (measured on an analytical balance) of the pellicles were determined prior to tensile testing, and all pieces were collected and freeze dried to determine the dry content and sample density. Data are presented as means ± SDs (both in tables and as error bars in bar charts) unless otherwise indicated. The data from tensile testing was statistically analyzed by the analysis of variance (ANOVA) single factor method with significant differences at $p < 0.05$. Additional details from the ANOVA analysis are provided in the Supplementary Information.

## Data availability
The authors declare that all relevant data are available within the article, the Supplementary Information or upon request from the corresponding authors. The data underlying Figs. 1d, 2a, 3c, 4 and Supplementary Fig. 7 are provided with this paper as a Source Data file.

## Code availability
The MATLAB code for the simulation of the poroelastic properties of the bacterial cellulose–hemicellulose hydrogels is available upon reasonable request from the corresponding authors.

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

## Acknowledgements

The Knut and Alice Wallenberg Foundation is acknowledged for funding through the Wallenberg Wood Science Center (WWSC). F.V. acknowledges the Swedish Research Council (VR) for financial support (project 621-2014-5295). G.Y. acknowledges the financial support from the Australian Research Council (project DP180101919) and the Biotechnology and Biological Sciences Research Council (BBSRC) (project BB/T006404/1). Dr. Shoaib Azhar is thanked for providing acGGM from thermomechanical pulping (TMP). Dr. Martin Lawoko and Dr. Jakob Wohlert (WWSC) are thanked for valuable discussions. We gratefully acknowledge Professor Jason R. Stokes (School of Chemical Engineering, University of Queensland) for providing access to the rotational rheometer and helpful discussions regarding rheological measurements, Dr. Grant Edwards (Australian Institute for Bioengineering and Nanotechnology, University of Queensland) for helpful discussions regarding the tensile testing, as well as SCA Munksund AB (Sweden) for providing wood material.

## Author contributions

J.B., M.J.G., and F.V. conceived the idea and D.M., G.E.Y., M.J.G., and F.V. supervised the research. J.B., D.M., B.M.F., S.D., and S.G. performed the experiments. J.B. and G.E.Y. developed the theoretical framework for mechanical data analysis. J.B., G.E.Y., M.J.G., and F.V. wrote the manuscript with input from D.M., G.H., and M.E.L., and comments from all the authors.

## Funding

## Competing interests

The authors declare no competing interests.
