## [Peer Review File · Nature Communications]

Reviewers' Comments:

Reviewer #1:

Remarks to the Author:

What are the major claims of the paper? Are the claims novel? If not, please identify the major papers that compromise novelty. Will the paper be of interest to others in the field? Will the paper influence thinking in the field?

The paper reports a thorough study of the effects of individual and binary mixtures of wood hemicelluloses on the mechanical properties of bacterial cellulose pellicles, i.e. non-woven mats of bacterial cellulose fibrils. The reported results demonstrate that wood mannans increase the compressive modulus of the pellicles, whereas wood xylans increase their tensile ductility. The major claims of the paper are that these effects are indicative of i) a here-to-fore unrecognized role of hemicelluloses in wood cell walls and ii) differing roles for mannans and xylans. While hemicelluloses have previously been recognized to play a role in the mechanical performance of wood cell walls, this role was believed to lie at the cellulose-lignin interface. Specifically, hemicelluloses are thought to, among other functions, mechanically couple lignin to cellulose microfibrils. The claims that hemicelluloses modulate the mechanical properties of a cellulose fibril network and that xylans and mannans have quasi-opposing effects in this regard are unprecedented. These novel claims will most certainly influence thinking in the field and are likely to guide future technology development. The paper will be of large interest to others in the field and holds the promise to catalyze the development of a new mechanical model for secondary plant cell walls.

Are the claims convincing? If not, what further evidence is needed? Are there other experiments that would strengthen the paper further? How much would they improve it, and how difficult are they likely to be?

The paper's claims are convincing. The authors have thoroughly evaluated their model systems by verifying that i) the hemicellulose phase is uniform throughout the cellulose fibril mat, ii) the sugar composition in the hemicellulose phase is representative of the hemicellulose(s) present, iii) the morphology and crystallinity of the cellulose fibrils are unaffected. In their justification of bacterial cellulose as a suitable model for non-lignified plant cell walls, the authors should briefly acknowledge the difference in crystal composition (contents of I α and I β allomorphs) of the two cellulose origins and discuss its relevance or lack thereof for the interpretation of results.

Computational modeling of hemicellulose adsorption onto different cellulose I α and I β crystallographic faces could provide supporting evidence for the assumption that bacterial cellulose fibrils are a valid model for plant cellulose microfibrils. However, such MD simulations are still pushing the limits of current computational methods, would significantly delay publication of these impactful experimental results, and would represent groundbreaking results on their own, needing to be discussed in detail, separately. Therefore, such evidence could not be reasonably required. The synthesis of hemicelluloses with well-defined molecular patterns is currently not technically feasible. Extracted hemicellulose, as used here, are currently our best option for studying cellulose-hemicellulose systems. The authors have provided detailed evidence in support of the stated molecular properties/structures of the extracted hemicelluloses.

Are the claims appropriately discussed in the context of previous literature?

The discussion of results demonstrates extensive knowledge and a deep understanding of plant cell wall and hemicellulose structure. One aspect that could have been made clearer is that the proposed model integrates current knowledge of hemicellulose-cellulose interactions, solution-state properties (e.g. chain flexibility) and aqueous solubility of hemicelluloses, and lignin carbohydrate complexes (presumably limiting elastic deformation) into one cohesive mechanical model. While the proposed model is clearly described, our current knowledge in regards to the

three relevant interfaces (hemicellulose/cellulose, hemicellulose/water, hemicellulose/lignin) could be more clearly summarized early on in the manuscript.

If the manuscript is unacceptable in its present form, does the study seem sufficiently promising that the authors should be encouraged to consider a resubmission in the future?

In some places, the discussion and explanations are somewhat wordy and difficult to follow. Minor revision of the manuscript is recommended to clarify explanations and, thus, the significance of the findings.

Also, the current title of the manuscript does not reflect the impact of the results. Bacterial cellulose hydrogels have very limited uses. A focus on "cellulose fibrils" in general would deemphasize the model system and enable experts in the field to recognize the impact of the report.

Report prepared by
Maren Roman

Reviewer #2:

Remarks to the Author:

As a model study, making pseudo-plant cell wall using in-situ addition of plant hemicellulose into the bacterial cellulose network is not a new approach and many reports have been published. The arguments on the applicability of such knowledge for creation of new biomaterial is still speculative.

Therefore, the manuscript would seem of considerable interest to those working in the similar area of plant science and the degree to which the results will stimulate discussion in the field.

Reviewer #3:

Remarks to the Author:

The manuscript by Vilaplana et al., "Wood Hemicelluloses Exert Distinct Biomechanical Contributions in Bacterial Cellulose Hydrogels" provides an interesting discussion related to the possible biomechanical roles of the major types of heteropolysaccharides found in secondary plant cell walls of hardwood and softwoods (mannans and xylans). This is a topic that has attracted recent fundamental attention in the scientific community where plant biologists are starting to reveal the possible effect of structural sequencing on the interactions with cellulose. These are fascinating aspects that deserve attention given their implications not only from the point of view of molecular and plant biology but material science and other disciplines.

The paper is generally well written and the discussion is supported by the experimental results, under tests that were designed appropriately. Some issues still deserve attention before further consideration:

The choice of microbial cellulose should be better justified. For instance, I would imagine that the use of model thin films of cellulose, either from wood or from microbial cellulose, could have been used to study the adsorption of the hemicelluloses. On the other hand, by incubating the system in the presence of the hemicelluloses interferes with a number of variables that make the system quite complex for analysis. For instance, the biosynthesis process itself can be affected in ways that are not as obvious. Hence, some of the mechanical properties can be explained in light of the effect that the polysaccharides have in bacteria growth. This has been shown with other polymers, including CMC, PEO, chitin and others. Would it have been even more appropriate to add the hemicelluloses after the pellicle is formed? Incubation in the presence of hemicellulose interferes with cellulose production and obscures the central aspect of understating the interactions. Thus, he

biomechanical response in cellulose/hemicellulose hydrogels can in principle be controlled by the changes produced in the synthesis process.

A critical point for discussion is that of intermolecular interactions and supramolecular arrangements and their effect on the biomechanical properties of the assemblies. The authors often refer to the type of conformations and interactions. However, I find it somewhat speculative, given the type of approaches used. For instance, I would have expected a more insightful and direct assessment if surface forces were evaluated, for instance, via AFM or SFA. QCM/SPR analyses could also add some additional information, related to binding.

The distinct biomechanical behaviors are ascribed to the presence of specific molecular features in xylans and mannans, affecting the supramolecular arrangement. However, evidence for this is not clear. Aspects such as the rigidity of the formed domains, adhesion and adsorption, the "loops and tails" formed and bridging phenomena are quite complex and assessed with tools that are not entirely appropriate.

Given the effects of local viscosity, I wonder if it is necessary to provide more details about electrokinetic aspects. For instance, what is the charge of the hemicelluloses used? (at the given pH and ionic strength used). What is the effect of the proteins and hemicellulose interactions? Other than the points above, I think that this paper is quite interesting and appropriate to the general readership of Nature Comm

Reviewers' Comments:

Reviewer #1:

Remarks to the Author:

The manuscript is much improved and deserves publication in Nature Communications. A few suggestions and considerations are given below.

Page 2, Line 54: The expression "structural nature of the interactions" is unclear. Would "structural impact of the interactions" be more descriptive?

Page 2, Line 59-60: I suggest changing "suggested to act as a link between the lignin and cellulose components in the cell wall, regulating the aggregation of cellulose microfibrils" to "suggested to act as a link between the lignin and cellulose components in the cell wall and to regulate the aggregation of cellulose microfibrils"

Page 2, Line 70: The expression "as barriers" is unclear. Would "as barrier films" be more descriptive?

Page 4, Line 125-127: The acronyms AGX, acGGM, and acGX were already defined on Page 2.

Page 7, Line 238: Erelax was already defined on line 236. Therefore, the sentence could start with "Erelax is computed".

I find Figure 5 somewhat misleading, unless I misunderstood the explanation. The use of the words motifs and patterned or non-patterned domains on Page 17 suggests that the hemicellulose molecules have segments (motifs) that tightly adhere to the microfibril surface and other segments that prefer to stay hydrated. Figure 5 (a) implies that there are two types of Birch acGX, a rigid, attached version and a flexible, detached version. In my mind, each molecule may have rigid and flexible segments and may be partially attached to the microfibril surface with loose ends, or free dangling loops (but see below), making up the remainder of the molecule. This model is consistent with a rigid and a flexible phase but does not assume a rigid hemicellulose fraction and a flexible hemicellulose fraction. I would have expected a drawing that shows bare xylan segments that adhere with adjacent glucuronic acid-decorated segments that dangle.

One aspect that might be worth mentioning is the estimated length of the hemicellulose molecules in nanometers to enable the reader to judge whether one hemicellulose molecule would be able to bridge two microfibrils or whether this is unlikely. If my textbooks are correct, hemicelluloses are between 70 and 200 residues long. Assuming a length of 1 nm per 2 residues gives an approximate molecular length of 35-100 nm. The small length of the hemicellulose molecules makes bridging and even looping unlikely, given the fairly low flexibility (high persistence length) of polysaccharides.

I agree with Figure 5 (b) in that dangling ends might be the most likely manifestation of the flexible phase. However, the length of the hemicellulose molecules in Figure 5 (b) seems misleading. A length of 35-100 nm, would be 4 to 12.5 times the thickness of the bacterial cellulose microfibrils (8 nm) and 1 to 3 times the width (35-50 nm). The image seems to be depicting the microfibrils in profile, i.e. the thickness of the blue line would correspond to 8 nm.

Knowing the weight fractions of microfibrils and hemicelluloses in the composites, the authors might be able to estimate whether there is a non-adsorbed hemicellulose fraction, as depicted in Figure 5, or whether the surface area of the bacterial cellulose fibrils, which can be estimated, would be large enough to provide an excess of adsorption sites, reducing the non-adsorbed fraction.

Maren Roman

Response to Reviewers

Manuscript **NCOMMS-20-02344B**

Title: “Wood Hemicelluloses Exert Distinct Biomechanical Contributions to Cellulose Fibrillar Networks”

We acknowledge the valuable contribution from Reviewer #1 for all the comments and suggestions. We have made the requested modifications to the text. The detailed response to the reviewers are listed below and the corrections are highlighted in blue in the main manuscript.

REVIEWERS' COMMENTS:

Reviewer #1 (Remarks to the Author):

The manuscript is much improved and deserves publication in Nature Communications. A few suggestions and considerations are given below.

Page 2, Line 54: The expression “structural nature of the interactions” is unclear. Would “structural impact of the interactions” be more descriptive?

We have changed the expression to “The chemical nature and structural impact of the interactions ...”

Page 2, Line 59-60: I suggest changing “suggested to act as a link between the lignin and cellulose components in the cell wall, regulating the aggregation of cellulose microfibrils” to “suggested to act as a link between the lignin and cellulose components in the cell wall and to regulate the aggregation of cellulose microfibrils”

We have changed the formulation of the sentence as proposed.

Page 2, Line 70: The expression “as barriers” is unclear. Would “as barrier films” be more descriptive?

We have added “barrier films” as suggested.

Page 4, Line 125-127: The acronyms AGX, acGGM, and acGX were already defined on Page 2.

The acronym definition has been removed from Page 4, as they were already introduced.

Page 7, Line 238: E_{relax} was already defined on line 236. Therefore, the sentence could start with “ E_{relax} is computed”.

The acronym definition for E_{relax} has been removed from Line 238.

I find Figure 5 somewhat misleading, unless I misunderstood the explanation. The use of the words motifs and patterned or non-patterned domains on Page 17 suggests that the hemicellulose molecules have segments (motifs) that tightly adhere to the microfibril surface and other segments that prefer to stay hydrated. Figure 5 (a) implies that there are two types of Birch acGX, a rigid, attached version and a flexible, detached version. In my mind, each molecule may have rigid and flexible segments and may be partially attached to the microfibril surface with loose ends, or free dangling loops (but see below), making up the remainder of the molecule. This model is consistent with a rigid and a flexible phase but does not assume a rigid hemicellulose fraction and a flexible hemicellulose fraction. I would have expected a drawing that shows bare xylan segments that adhere with adjacent glucuronic acid-decorated segments that dangle.

This is a very important reflection and we really appreciate the reviewer for raising this issue. Indeed, as the reviewer says, our intention was to represent that the same hemicellulose molecule could have both rigid and flexible domains, which both could be partially attached to the microfibrils and could have loose/dangling ends (e.g. see the loose ends arising from some of the original acGX, acGGM and AGX molecules). However, the presence of decorations (e.g. glucuronic acid) would not hinder in specific cases the adsorption of the hemicelluloses to the cellulose surfaces. As we know from recent studies (and also described in the text), the presence of patterned motifs in glucuronoxylans (acGX and AGX) with even placement of decorations, enables interactions with cellulose microfibrils.^{1, 2, 3, 4, 5} We have therefore drawn these regular motifs with even placement of substitutions directly interacting with cellulose surfaces, whereas the minor acGX and AGX motifs with uneven and consecutive decorations^{5, 6} are drawn as dangling/hydrated segments. On the other hand, there is indirect evidence that the presence of consecutive Glc-Man motifs in glucomannans would as well enable their close interactions with cellulose surfaces.^{7, 8} Therefore, we have drawn preferentially the segments in acGGM with higher Glc content and consecutive Glc-Man sequences in direct contact with the cellulose surfaces, whereas the other potential motifs are displayed in hydrated loops.

As we can assume from the reviewer's comment, this was not clearly portrayed in Figure 5a. We have simplified and redrawn Figure 5a extending the length of the depicted hemicellulose molecules and portraying both the presence of rigid and flexible conformations in the same hemicellulose molecule. We have added as well a description in the Figure 5 legend hopefully clarifying this issue.

One aspect that might be worth mentioning is the estimated length of the hemicellulose molecules in nanometers to enable the reader to judge whether one hemicellulose molecule would be able to bridge two microfibrils or whether this is unlikely. If my textbooks are correct, hemicelluloses are between 70 and 200 residues long. Assuming a length of 1 nm per 2 residues gives an approximate molecular length of 35-100 nm. The small length of the hemicellulose molecules makes bridging and even looping unlikely, given the fairly low flexibility (high persistence length) of polysaccharides.

We performed size exclusion chromatography on the isolated wood hemicelluloses (Supplementary Table 1) showing average molar masses between 20 – 40 kDa, which indeed correspond with a degree of polymerization of 100 – 200. The assumption of 1 nm per 2 residues provides a theoretical length of a perfectly elongated hemicellulose chain, corresponding to 50-100 nm. This molecular size for hemicelluloses is an overestimation, as hemicelluloses are not strictly linear biopolymers (since they have decorations) and they are not strictly rigid rods (since their flexibility is modulated by the presence of decorations). In our recent studies, we provide values of the hydrodynamic radius in the order of 10 nm for spruce glucomannan and xylan molecularly dissolved in water, where the hemicelluloses probably adopt more compact random coiled conformations.^{9, 10} Moreover, hemicelluloses have large tendency to self-aggregate in hydrated conditions into larger fractal organizations that can reach 100 – 1000 nm.^{9, 10} We consider that the coiled and extended conformations constitute the two extreme cases for the real situation, as individual hemicellulose molecules can adopt different conformations attached to the cellulose microfibril surfaces and in hydrated/solution state.^{2, 4, 5, 11} This explanation has been added in the manuscript (page 5).

With this in mind, we can only speculate with the possibility if a secondary cell wall hemicellulose could bridge two individual cellulose microfibrils. Based on an extended macromolecular size of the hemicellulose, the bridging ability of hemicelluloses could be theoretically possible in regions where cellulose microfibrils are close to each other, but this seems improbable due to hemicellulose chain stiffness and molecular dynamics. Indeed, the tethering model for hemicelluloses (in this case, xyloglucans) in primary plant cell walls has been recently revised by a 'biomechanical hotspots' model, where xyloglucan can mediate potential cellulose–cellulose contacts.¹² As our study cannot provide such information at the individual macromolecule level, we propose a fibrillar architecture as depicted in Figure 5b where the hemicelluloses are extended in rigid phases directly interacting with the cellulose surfaces, together with flexible phases with more coiled/hydrated conformations where hemicelluloses can interact with each other through bridging adhesion of different intensities (strong in the case of glucomannans, weaker in the case of xylans). This organization of hemicelluloses in rigid and flexible phases would in turn modulate the interactions between hemicellulose-coated cellulose microfibrils and the resulting biomechanical properties. The implementation of coarse-grained simulations with this multiphase system would definitely enable to test the different organizational hypotheses at the individual macromolecule level, and their effect of the resulting biomechanical properties. Although we prefer not to include the discussion of the potential bridging capacity of hemicelluloses in the manuscript, we have clarified the potential conformations of the hemicelluloses in the rigid/flexible phases in the caption to Figure 5b.

I agree with Figure 5 (b) in that dangling ends might be the most likely manifestation of the flexible phase. However, the length of the hemicellulose molecules in Figure 5 (b) seems misleading. A length of 35-100 nm, would be 4 to 12.5 times the thickness of the bacterial cellulose microfibrils (8 nm) and 1 to 3 times the width (35-50 nm). The image seems to be depicting the microfibrils in profile, i.e. the thickness of the blue line would correspond to 8 nm.

We appreciate this insightful comment. We have modified the Figure 5b so the dimensions of the bacterial cellulose microfibrils and the hemicelluloses are in a more realistic scale.

Knowing the weight fractions of microfibrils and hemicelluloses in the composites, the authors might be able to estimate whether there is a non-adsorbed hemicellulose fraction, as depicted in Figure 5, or whether the surface area of the bacterial cellulose fibrils, which can be estimated, would be large enough to provide an excess of adsorption sites, reducing the non-adsorbed fraction.

In the manuscript we already estimated the ratio of flexible non-adsorbed hemicelluloses using Solid-state ¹³C NMR analysis, where around 50-60 % of incorporated hemicelluloses exhibited rigid interaction with cellulose. For clarification, we have added this information in the Caption to Figure 5b. As the composition of hemicelluloses in the BC hydrogels is around 20-25 wt% we can already anticipate that the cellulose microfibrils provide an excess of adsorption sites that are not occupied by the hemicelluloses.

Maren Roman

References

1. Grantham NJ, *et al.* An even pattern of xylan substitution is critical for interaction with cellulose in plant cell walls. *Nature Plants* **3**, 859-865 (2017).
2. Simmons TJ, *et al.* Folding of xylan onto cellulose fibrils in plant cell walls revealed by solid-state NMR. *Nature Communications* **7**, 13902 (2016).
3. Busse-Wicher M, *et al.* Evolution of Xylan Substitution Patterns in Gymnosperms and Angiosperms: Implications for Xylan Interaction with Cellulose. *Plant Physiology* **171**, 2418-2431 (2016).

4. Busse-Wicher M, *et al.* The pattern of xylan acetylation suggests xylan may interact with cellulose microfibrils as a twofold helical screw in the secondary plant cell wall of *Arabidopsis thaliana*. *The Plant Journal* **79**, 492-506 (2014).
5. Martínez-Abad A, *et al.* Regular Motifs in Xylan Modulate Molecular Flexibility and Interactions with Cellulose Surfaces. *Plant Physiology* **175**, 1579-1592 (2017).
6. Martínez-Abad A, Giummarella N, Lawoko M, Vilaplana F. Differences in extractability under subcritical water reveal interconnected hemicellulose and lignin recalcitrance in birch hardwoods. *Green Chemistry* **20**, 2534-2546 (2018).
7. Martínez-Abad A, Jiménez-Quero A, Wohler J, Vilaplana F. Influence of the molecular motifs of mannan and xylan populations on their recalcitrance and organization in spruce softwoods. *Green Chemistry* **22**, 3956-3970 (2020).
8. Yu L, *et al.* The Patterned Structure of Galactoglucomannan Suggests It May Bind to Cellulose in Seed Mucilage. *Plant Physiology* **178**, 1011-1026 (2018).
9. Kishani S, *et al.* Experimental and Theoretical Evaluation of the Solubility/Insolubility of Spruce Xylan (Arabino Glucuronoxylan). *Biomacromolecules* **20**, 1263-1270 (2019).
10. Kishani S, Vilaplana F, Xu W, Xu C, Wägberg L. Solubility of Softwood Hemicelluloses. *Biomacromolecules* **19**, 1245-1255 (2018).
11. Zheng Y, Wang X, Chen Y, Wagner E, Cosgrove DJ. Xyloglucan in the primary cell wall: assessment by FESEM, selective enzyme digestions and nanogold affinity tags. *The Plant Journal* **93**, 211-226 (2018).
12. Cosgrove DJ. Re-constructing our models of cellulose and primary cell wall assembly. *Current Opinion in Plant Biology* **22**, 122-131 (2014).